

# Planform river channel perturbations resulting from active landsliding in the High Himalaya of Bhutan

Larissa de Palézieux[1], Kerry Leith[1], and Simon Loew[1]

[1]Department of Earth Sciences, Engineering Geology, ETH Zurich, CH-8092 Zurich, Switzerland

**Correspondence:** Larissa de Palézieux (larissa.depalezieux@erdw.ethz.ch)

**Abstract.** Large creeping landslides are persistent features in mountainous landscapes. Evaluating the long-term evolution of these features and associated present-day hazards is however difficult. We use a Fourier transform to characterize planform channel sinuosity and find that the amplitude at given wavelengths follows the power law of pink noise ($1/f_{noise}^a$) with an exponent of $a = 1.1$, which is consistent with a fractal distribution. This allows us to distinguish local landslide perturbations from the background sinuosity of the unperturbed channels. In order to quantify the interaction of landslides with river channels, we use a new metric for landslide-induced channel offset, which allows us to identify exceptional amplitudes associated with landslide activity. We find that 83% of the 226 mapped large creeping landslides in the High Himalaya of Bhutan have generated lateral channel migration in the direction of the landslide displacement. Assuming landslide initiation is associated with knickpoint propagation, our derived stream power normalized rates of landslide-induced channel offset range from 2 $\cdot 10^{-1}$ to 2 $\cdot 10^{-2}$ m$^{-0.9}$. These rates are consistent with an early period of relatively rapid landslide displacement followed by a long period of stabilization, and finally, a gradual acceleration of more mature landslides. Assuming constant bedrock erodibility, displacement rates derived from the landslides in our study region may provide inside into the evolution of large creeping landslides over a period of 1 Myr.

## 1 Introduction

### 1.1 Background and Motivation

Large ($> 1$ Mm$^3$) creeping landslides generate progressive topographic change over 1 kyr to 1 Myr at rates of $\sim 1$ mm yr$^{-1}$ (Hungr et al., 2014). Even though the probability of associated catastrophic failure is low (Hendron Jr and Patton, 1985; Intrieri et al., 2018; Lacroix et al., 2020), insufficient consideration of risks associated with rock mass degradation, ongoing creep, and parasitic failures can have serious consequences (Hendron Jr and Patton, 1985; Bonzanigo et al., 2007; Dorji, 2019). Sensitive (human) mountain infrastructure like hydro power dams are commonly constructed in narrow valley sections so as to reduce the amount of material required and to lessen the dam stability requirements. A careful consideration of processes driving geomorphological changes at such sites prior to construction can provide important insight into the mechanical behaviour of the local sub-surface, predict changes in stress, and minimize the risk of natural hazards affecting both the dam and construction workers. Here, we propose an approach combining methods from classical fluvial geomorphology, as well as a new method





to assess channel displacements due to landsliding. Together these methods allow us to identify zones of heightened erosional activity in a large area ($10^4$ km$^2$) and to characterize the activity and evolutionary stage (and by proxy degree of rock mass degradation) at individual landslide locations.

Modification of bedrock channels by fluvial processes is the result of uplift and erosional processes (Whipple, 2004; Perron and Royden, 2012; Willett et al., 2014), and is commonly assumed to be disconnected from those governing the channel-

adjacent hillslopes (Li et al., 2020). However, hillslope processes, such as landsliding, affect both bedrock strength and sediment transport and therefore contribute to channel erosion with respect to sediment supply and bedrock erodibility (Hovius et al., 1997; Korup et al., 2010; Larsen et al., 2010). Research investigating the long-term evolution of stream networks typically employs estimates of regional erodibility, stream power, and tectonic uplift, to evaluate the present-day erosional activity and likely future evolution of longitudinal river profile (Whipple, 2004; Perron and Royden, 2012; Willett et al., 2014). In

this context, erodibility is back-calculated from the assumed steady-state profile, and is a cluster term accounting for bedrock strength properties and erosional efficiency of the fluvial system. By incorporating information regarding the spatial distribution of ongoing erosional processes both along the river profile and on the adjacent hillslopes, the relationship between the evolution of stream networks and hillslopes can be studied in more detail. This provides additional insight into the long-term evolution of the stream network and allows us to contextualise the spatial and temporal distribution of hillslope processes, and

in particular landslides, in an engineering geological framework.

In this study, we use channel planform morphology to derive normalized rates of channel offset due to fluvial incision and landsliding. We combine methods from fluvial geomorphology and signal processing to develop metrics that allow us to quantify the deviation of river channels in locations of mapped slope instabilities and compare these offsets to the deviations found for the stream network.

## 1.2  Characteristic river and hillslope features

In order to evaluate the interaction between large creeping landslides and river channels, it is important to first understand the processes that lead to the development of characteristic fluvial and hillslope morphologies.

Rivers exert a first-order control on the evolution of landscapes in tectonically active regions (Willett, 1999; Whipple, 2004). The rate of relief production is limited by detachment-limited fluvial erosion at the valley axis, while empirical equations

tying sediment size to erosion rate indicate that transport and redistribution of sediment within a drainage basin determines the efficiency of sediment evacuation from active hillslopes (Li et al., 2020). Rates of sediment comminution affect both spatial and temporal erosion patterns. Ultimately, evacuation of weathering and erosion products from a landscape determines the long-term mass balance of tectonic orogens. Many of the aspects regarding sediment production and evacuation can be evaluated by studying the morphology of longitudinal river profiles and provide a basis with which we can begin to assess regional erosional

activity. River profiles consistently demonstrate an exponential increase of slope with decreasing drainage area (Hack, 1957). Drainage area is therefore regularly used as a proxy for river discharge (Wobus et al., 2006). Perturbations to these exponential profiles are commonly associated with convex breaks in river profile slope such as knickpoints or waterfalls, where the river channel runs on bedrock with a increase in turbulent flow due to the locally increased profile slope (Whipple and Tucker, 1999;



Earth **Surface**
Dynamics
Discussions

Royden and Perron, 2013). Empirical relationships such as the stream power law (Whipple and Tucker, 1999; Whipple, 2004),

have been shown to be capable of broadly reproducing characteristics of longitudinal river profiles (including 'channel slope' - 'catchment area' relationships and knickpoint distributions) by incorporating the effects of steady-state, or variable tectonic uplift and climate at timescales relevant to relief production in alpine regions (Fox et al., 2015; Leith et al., 2018). The stream power law itself expresses erosion rate $dz/dt$ as function of drainage area $A$, erodibility of the bedrock $K$, channel slope $S$, and parameters for uplift and erosion ($m, n$):

$$\frac{dz}{dt} = E = K A^m S^n \tag{1}$$

where $m$ and $n$ are integers $\in [0, 1]$ and typically chosen such that the concavity index $\Theta = m/n = 0.45$.

From this, channel slope can be written as

$$S = k_s A^{-\Theta} \tag{2}$$

with

$$k_s = \left( \frac{U}{K} \right)^{1/n} \tag{3}$$

where $k_s$ is the steepness index and $U$ and $K$ uplift rate and erosional efficiency respectively.

The longitudinal river profile is described by solving the stream power equation for steady-state conditions (Perron and Royden, 2012):

$$z(x) = z_b + \left( \frac{U}{K A_0^m} \right)^{\frac{1}{n}} \chi \tag{4}$$

with

$$\chi = \int_{x_b}^{x} \left( \frac{A_0}{A(x')} \right)^{\frac{m}{n}} dx' \tag{5}$$

where $z_b$ is the base level elevation [m a.s.l.], $U$ is the uplift rate [m a$^{-1}$], $K$ is the erosional efficiency [m$^{1-1m}$ a$^{-1}$], and $A_0$ is the reference drainage area [m$^2$] which is typically set to 1 m$^2$.

The formation of knickpoints and their subsequently upstream propagation can drive the development of landslides on

adjacent hillslopes (Hou et al., 2014; Tsou et al., 2014). Knickpoints form in response to a discrete decrease in relative uplift rate, and / or increase in erosion rate, and propagate upstream as a kinematic wave whose speed is related to a combination of streampower and bedrock resistance (Berlin and Anderson, 2007; Loget and Van Den Driessche, 2009). A hillslope adjacent to this stream network may be expected to experience a reduction in global stability in the wake of the passing knickpoint





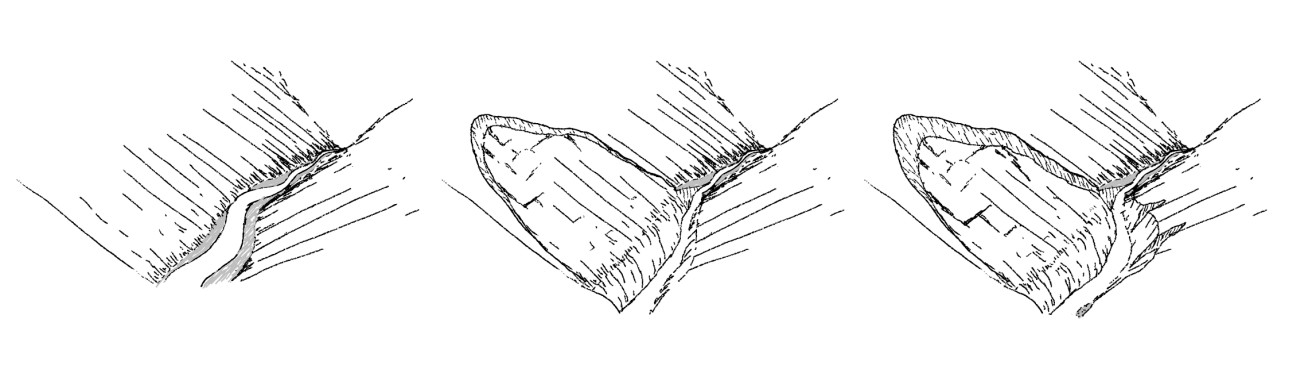

**Figure 1.** Schematic illustration of the progressive failure of a channel-adjacent hillslope leading to the formation of a landslide and the subsequent lateral displacement of the channel alignment: a) stable fluvial channel in a steady-state landscape, b) a wave of incision initiates a creeping rock slope failure as a knickpoint passes to the upstream extent, c) continued landslide displacement degrades the rock mass, stalls knickpoint migration, and displaces the channel toward the opposite side of the valley.

as incision lowers the local baselevel, increasing local relief and therefore driving stress and/or exposing buried planes of
weakness (Fig. 1) (Tsou et al., 2014; Korup and Schlunegger, 2007; Korup et al., 2010).

    The mobilization of rock masses, and subsequent landslide development occurs when the driving forces (typically resulting from gravity or seismicity) outweigh the resisting forces (from material strength and external support) (Terzaghi, 1962; Eberhardt et al., 2004). Fluvial incision directly impacts both the driving and the resisting forces by increasing local relief and removing external support at the slope toe. While a reduction in global stability resulting from incision may be insufficient to
immediately initiate slope displacement, a relative increase in the local stress/strength ratio can markedly increase the rate of damage propagation within the hillslope, and retrogressive or compound landslides may initiate several thousand years after the initial perturbation (the European Alps, for example, appears to have experienced an increase in landslide frequency 2-3 kyr, and 8-9 kyr after deglaciation (Prager et al., 2008)).

    This transitional stage of oversteepened hillslopes is intuitively captured in the concept of excess topography by Blöthe
et al. (2015), who quantify the amount of topography in excess of a mechanical threshold, defined by the internal friction angle of the rock mass: hillslopes with angles higher than the internal friction angle are assumed to be in static disequilibrium. The part of the topography 'in excess' of the internal friction angle in this manner is then termed excess topography and assumed to be subject to future erosion as a result of the progressive decrease of rock mass strength as internal fracture propagation and weathering decreases cohesion. Calculating excess topography provides a snapshot of the regional state of
erosional equilibrium: zones with low or no excess topography are likely to be in equilibrium with the erosional state of the stream network, while zones with higher excess topography are in disequilibrium and may tend towards re-equilibration through mass wasting processes.





Similar to longitudinal river profiles, the planform morphology of stream networks is also a result of a combination of channel wall strength and erosional processes. While the effects of vertical incision have been studied in some detail (Whipple, 2004; Willett, 1999; Turowski et al., 2008), less attention has been paid to lateral erosion (Turowski et al., 2008; Li et al., 2020). In order to investigate the directional components of erosion and their respective contribution to total erosion, Li et al. (2020) developed an emprirical model for lateral erosion by bedload particle impacts and coupled it with the saltation-abrasion vertical erosion model (Sklar and Dietrich, 2004). They determine that the erosion rate ($E_c$) for the channel walls is a function of bedrock strength, channel armouring, and particle impact energy:

$$E_c = I_w \frac{\pi \rho_s D^3 v_i^2 Y}{6 k_v \sigma_T^2} \tag{6}$$

where $I_w$ is the number of impacts of bedload particles on the channel wall per unit time, $\rho_s$ the sediment density, $D$ the diameter of roughness elements on the channel floor, $v_i$ the impact velocity of the bedload particles on the channel wall, $Y$ the Young's modulus of the bedrock, $k_v$ a dimensionless bedrock strength coefficient, and $\sigma_T$ the tensile yield strength.

Similar to the uplift rate contribution to the profile of fluvial channels, lateral impingement of creeping landslides should theoretically affect the planform of river channels. Lateral migration of a valley axis (as opposed to erosion described in Equation 6) may be considered to be the product of a long-term imbalance in lateral erosion, and/or relative displacement of one of the adjacent hillslopes while maintaining balanced erosion of the channel walls. In a case in which one side of a channel is continually displaced across the valley axis (e.g. through landsliding), the channel will need to erode slope material in order to maintain the cross-sectional channel geometry (i.e. channel width). The total (apparent) displacement on the side of the landslide $D_L$, i.e. the channel offset, will be equal to the absolute offset of the landslide $D_{abs}$ minus the erosion occurring at the landslide toe $E_{c,L}$:

$$D_L = D_{abs} - E_{c,L} \tag{7}$$

Assuming that the channel width will be maintained, the resulting offset of the channel wall will be limited by the erosion of the opposing bank $E_{c,S}$:

$$D_L = E_{c,S} \tag{8}$$

The rate of lateral erosion may be affected by the presence of creeping landslides and river banks in a number of ways. Previous landslide activity may specifically reduce the dimensionless bedrock strength coefficient $k_v$ the tensile yield strength $\sigma_T$ and increase rock mass density $\rho_s$ through enhanced weathering, while also increasing the mean diameter of roughness





elements by introducing coarse rockfall and rockslide debris to the channel. Assuming that the number of impacts $I_w$ and the
impact velocity $v_i$ average out along the toe of the landslide, we can compare the lateral erosion rate of the two sides:

$$\frac{E_{c,L}}{E_{c,S}} = \frac{\rho_{s,L} D_L^3 Y_L k_{v,S} \sigma_{T,S}^2}{\rho_{s,S} D_S^3 Y_S k_{v,L} \sigma_{T,L}^2} = \frac{\alpha_\rho \alpha_D^3 \alpha_Y}{\alpha_k \alpha_\sigma^2} \qquad (9)$$

where subscript $L$ denotes landslide and $S$ denotes stable channel wall, $\alpha_\rho = \rho_{s,L}/\rho_{s,S} > 1$, $\alpha_D = D_L/D_S > 1$, $\alpha_Y = Y_L/Y_S < 1$, $\alpha_k = k_{v,L}/k_{v,S} < 1$, and $\alpha_\sigma = \sigma_{T,L}/\sigma_{T,S} < 1$. Under the given assumptions the numerator of Eq. 9 will tend
to increase with respect to the denominator, suggesting an increased erosion rate on the side of the channel adjacent to the
landslide with respect to the erosion rate of the stable slope, i.e. $E_{c,L}/E_{c,S} > 1$.

The interaction of large creeping landslides and adjacent channels will fundamentally be controlled by the production of
sediment from the unstable hillslope, and evacuation from the toe of the slope by the adjacent river. The comminution of
grains, and transport of sediment from the site, consume stream power that would otherwise be available to incise the river
channel into bedrock. A landslide-generated increase in either sediment size, or volume may then be expected to reduce the
rate of incision at a location, and would require a local increase in stream power (e.g. through an increase in channel gradient)
to maintain complete sediment evacuation.

Based on our present-day knowledge of landslide formation and the evolution of stream networks both along-profile and
in planform, we expect the study of the distribution of landslides with respect to morphological markers of past, ongoing,
and likely future erosional activity to provide valuable insight into the nature of the interaction between stream networks and
their adjacent hillslopes. We compare metrics of fluvial morphology and patterns of hillslope characteristics to geomorphic
domains identified in studies of soil types in Bhutan and provide a method to analyse the long-term regional development of
landslides with respect to the stream network. This allows us to extract metrics of long-term landslide displacement, quantify
the interaction between landslides and the adjacent stream network, and ultimately to show that the landslide distribution
co-evolves with the stream network.

## 1.3 Geomorphological drivers of slope instability in NW Bhutan

In order to evaluate the manner in which large creeping landslides are coupled to the evolution of alpine river systems, we study
two large drainage basins in the northwest of Bhutan (Fig. 2). The region is currently undergoing a significant development of
hydro power capacity, and our results will contribute to the evaluation of pre-existing slope stability at hydro power sites in
the area. A detailed catalogue of large landslides in these drainage basins (Dini et al., 2020) provides a useful opportunity to
assess both the profile and the planform morphology of the regional stream network and several distinct geomorphic domains
with respect to slope instabilities.

The roughly N-S convergent tectonic setting of the continent-continent collision between the Indian and the Eurasian con-
tinental plates has generated a complex system of thrust faults and associated uplift which results in the Himalayan mountain
range (Heim and Gansser, 1939; Gansser, 1983). At the longitudes of Bhutan, the southern boundary of the mountain range
is delimited by a sequence of four faults: the Main Frontal thrust (MFT), the Main Boundary thrust (MBT), the Main Central



thrust (MCT), and the South Tibetan detachment (Heim and Gansser, 1939; Gansser, 1983; Adams et al., 2016) (see Fig. 2 for location of the MFT). There are, however, no known active fault traces mapped in NW Bhutan (Grujic et al., 2018). This apparent lack of active tectonic features may, in part, be a result of a paucity of geological mapping, though it is consistent with the flat ramp transition in W Bhutan (Le Roux-Mallouf et al., 2015) and is at least in part supported by a study of recent

seismicity locating just one major event (Mw = 8, 1714 A.D.) in Bhutan (Hetényi et al., 2018).

Four main tectonostratigraphic zones occur in Bhutan: the Siwalik group is composed of 'Miocene to Pliocene synorogenic deposits' and is separated from the overlying Lesser Himalayan zone by the MBT. The Lesser Himalayan zone is separated from the overlying Greater Himalayan zone by the MCT, which in turn is separated from the overlying Tethyan Himalayan zone by the STD. All three zones (Lesser Himalaya, Greater Himalaya, and Tethyan Himalaya) are composed of strongly

metamorphosed pre-Himalayan metasedimentary rocks (Long et al., 2012; Tobgay et al., 2012).

Two major drainage basins are delineated within our study area, with Paro to the East and Punakha to the West (Fig. 2 and 3). Within each basin five major geomorphological domains have been differentiated based on soil composition (Norbu et al., 2003; Portenga et al., 2015) between the southern and the northern border of the country, which follows the northernmost crest of the Himalayan range. Relief abruptly starts at the range front (Foothills (FH)) in both basins, while the steepness of the river

profiles only increases 50-70 km further north in the Southern Valleys (SV). The SV are characterized by densely vegetated fluvial valleys and gorges with steep slopes and frequent landsliding. At this location the elevation range of relief increases to span almost 4 km vertically. This is due to the coexistence of the deeply incised Inner Valleys and the high-elevation, low-relief surfaces. North of this zone, local relief reduces to approximately 2 km and – after a stretch of constant elevation in the Inner Valleys (IV) – rises in elevation at the same slope as the river profiles (Northern Valleys (NV)). Relief increases

again in the High Himalayan Plateau (HHP) with low-relief, glacially overprinted U-shaped valleys, and the High Himalayan Peaks (HHPe), where maximum elevation is reached at over 7000 m a.s.l. (Norbu et al., 2003; Portenga et al., 2015). The low-relief surfaces in SV and IV are partially glacierized (Adams et al., 2016), eroded at significantly lower rates than the steeper surrounding slopes (Grujic et al., 2006; Portenga et al., 2015; Adams et al., 2016), and are interpreted to be uplifted paleo-landscapes (Duncan et al., 2003; Grujic et al., 2006; Adams et al., 2015; Portenga et al., 2015).

Little data exists with respect to erosion rates in Bhutan. Portenga et al. (2015) measured erosion rates $0.96 \pm 0.16$ mm yr$^{-1}$ in the South, $0.70 \pm 0.06$ mm yr$^{-1}$ in the North, and 0.4 mm yr$^{-1}$ in the uplifted, relict surfaces. In the west of the country the authors measured exhumation rates of 1.4 mm yr$^{-1}$.

The distribution of the mean annual precipitation (MAP) within Western Bhutan is inversely correlated with increasing topography from south to north. MAP in the foothills is 3500 mm, which decreases to ∼300 mm in the Inner Valleys, and

∼150 mm in the High Himalayan Plateau. Of the 3500 mm MAP at the range front ∼75 % of it fall during the Indian summer monsoon (June-August). The Shillong plateau – ∼ 100 km to the South of the range front – casts a rain shadow on Eastern Bhutan reducing the MAP at the range front by about 50 % with respect to Western Bhutan (Bookhagen et al., 2005; Fick and Hijmans, 2017).

Large landslides in the study region were recently mapped by Dini et al. (2020) using optical satellite images and a high-

resolution DEM (©GoogleEarth; ALOS World3D, 5m Ground Sample Distance, (JAXA, 2017)). The resulting inventory of 912

Earth **Surface**
**Dynamics**
Discussions

**Figure 2.** Map of the study area colored by excess topography with the landslide inventory (Dini et al., 2020) and knickpoints. The stream network is colored by $k_{sn}$. Regions with high excess topography (brown) coincide with high values of $k_{sn}$ along the stream network and are delimited by a clustering of knickpoints at their upstream limits. Excess topography and the hillshade were calculated based on two DEMs (JAXA, 2017; NASA JPL, 2013). *Situation inset:* MFT refers to the main frontal thrust (Valli, 2005).





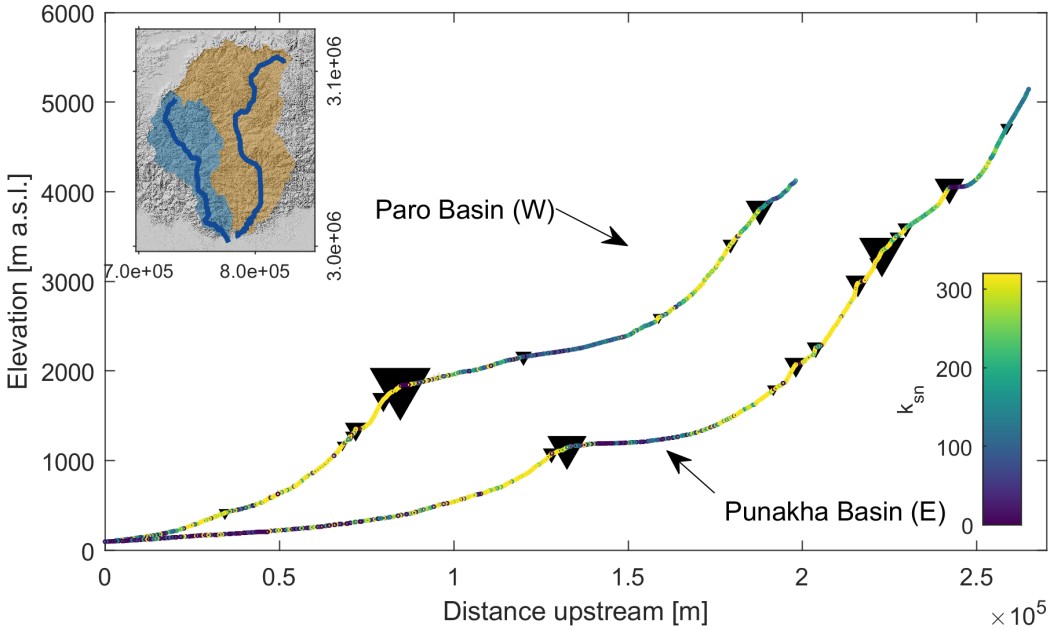

**Figure 3.** Swath profile of the two major basins (Paro (W) and Punakha (E)) with the trunk channel colored by $k_{sn}$ and the knickpoints scaled by vertical offset to the idealized profile.

features covers northwestern Bhutan, approximately between 27.3°and 28.0°N and 89.0°and 90.0°and includes, a qualitative classification for landslide activity based on observed phenomena ("increased rock fall activity with respect to surrounding, erosion at the toe, and deep and active debris flow channels"). Most landslides (628) were classified to have low activity (dormant or relict), 137 to be medium active (creeping), and 35 highly active (creeping and sliding; 112 not classified). The likelihood of the objects having been correctly identified as landslides is expressed in a semi-quantitative score composed of factors representing phenomena, which are typically expected to be observed in each of the classified gravitational landforms (Fig. 4). Roughly two thirds of the inventory were classified, with 265 of the landslides as uncertain and 316 from 'Possible' to 'Certain'.

The inventory comprises mostly large rock slope instabilities with surface areas between $5.6 \cdot 10^3$ m$^2$ and $1.6 \cdot 10^7$ m$^2$ and elevation ranges up to 1800 m. About half of the landslides in the inventory are located in the immediate vicinity of the stream network (at less than 50 m distance), almost all of the landslides are at most 0.5 km away from the channel, while less than 15% are at a distance of 0.5-3 km. A striking aspect of the slope instabilities of NW Bhutan is the almost complete lack of landslide deposits from catastrophic failures: in the field almost no landslide deposits can be seen at ground surface, suggesting that they have either been evacuated by erosion or buried by sediment deposition.

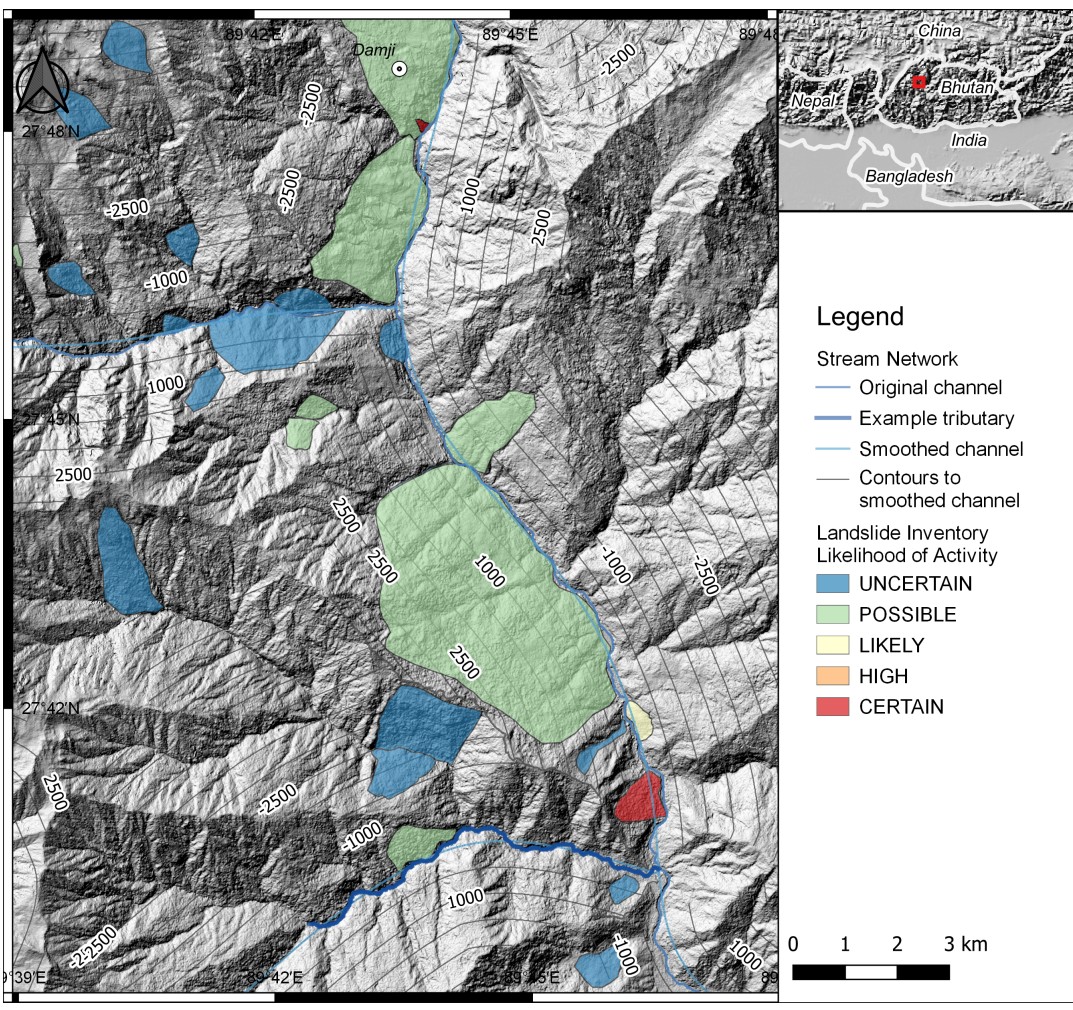

**Figure 4.** Overview of typical landslide-channel interactions with the stream network, landslide inventory (Dini et al., 2020) colored by likelihood of activity, and the major knickpoints. Contours illustrate the distance distribution to the valley axis. The example channel reach of Fig. 5 is highlighted in bold. The hillshade was derived from (JAXA, 2017)





## 2 Methods


In this study we quantify the interaction creeping landslides and river channels by correlating apparent channel displacement at the toe of landslides with atypical increases in channel sinuosity. We then investigate the rate of channel offset by evaluating relative rates of local knickpoint retreat. We define sinuosity as the amplitude of planform channel fluctuations relative to a spatial reference frame defined by the associated valley axis. Strong variations are then determined by considering the

amplitude of deviations at the location of mapped landslides with respect to typical sinuosities for similar half-wavelengths throughout the study region. The latter is determined by using a Fourier transform to determine characteristic amplitudes at all half-wavelengths ranging from that of the DEM resolution to $\sim 10$ km. Finally, we investigate the potential for a spatio-temporal relationship between landslide-induced increases in sinuosity and flow network evolution by associating landslides to large knickpoints identified upstream, and then comparing the apparent magnitude of sinuosity increase to 'time' since the

passing of the knickpoint assuming a constant bedrock erodbility and normalizing distance by upstream drainage area ($\chi$).

### 2.1 River channel network morphology

The river channel network used in this study was derived from an 11.9m DEM using a flow routing algorithm (Schwanghart and Scherler, 2014) with a minimum drainage basin area of 2.8 km$^2$, a minimum reach length of 10 km, and a minimum stream order of 2. The DEM was created by resampling a 5m ALOS DEM (JAXA, 2017) and completing it with resampled 30

m SRTM V3 data (NASA JPL, 2013).

We derive planform channel deviations, by extracting the deviation of the channel from a reference valley axis. The reference valley axis is generated from the channel network by smoothing each segment with a regularization method considering 1 km intervals and a stiffness penalty of 500 [−]. By calculating distance away from the valley axis, we determine a reference grid that allows us to re-plot channel position with respect to the valley axis (Fig. 5). The amplitude of sinuosity $D_{va}$ then reflects

fluctuations in the river channel toward, or away from, the valley axis.

In order to evaluate typical amplitudes of sinuosity within the study region, we calculate a Fourier transform with a sampling frequency of 1/DEM-cellsize. The output of the Fourier transform is an amplitude ($D_F$) distribution for the full range of given half-wavelengths per channel segment. Aggregating this information for all channel segments within the stream network yields a characteristic distribution of amplitude and half-wavelength for the entire stream network.

Knickpoints were extracted along the stream network using the knickpointfinder function of TopoToolbox (Schwanghart and Scherler, 2017). It extracts locations along the stream network which deviate vertically from a smooth and perfectly concave channel profile. The identification tolerance refers to the final vertical offset between the true and idealized profiles at the location of the knickpoint and is scaled to the uncertainty of the DEM. An uncertainty of 10-20 m was determined from the interquantile range of the stream network as outlined in Schwanghart and Scherler (2017) and the number of identified

knickpoints increased at tolerances above 100. We therefore conservatively set the tolerance to 100 m to avoid mis-identifying knickpoints due to DEM noise.





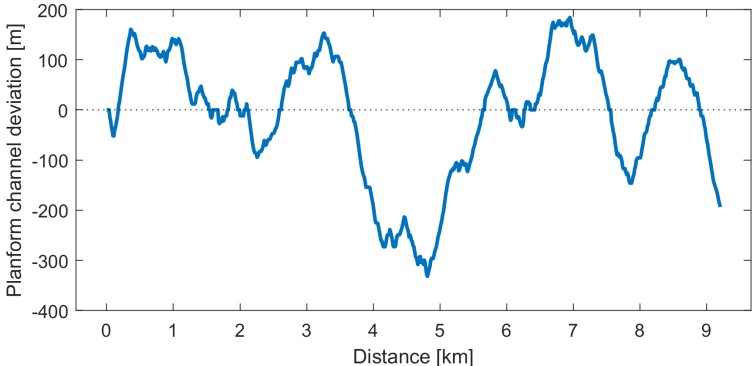

**Figure 5.** Amplitude of channel sinuosity $D_{va}$ with respect to the valley axis for one example reach. For location refer to Fig. 4.

## 2.2 Planform morphology of landslide toe region

In order to quantify the potential landslide-induced local offset of valley axes, we focus on the lower $1/3$ of the landslide (i.e., the accumulation region near the toe) and compare proximity distributions in the landslide referenced a) to the local channel and b) to the local valley axis. Both distributions are then normalized by the minimum value in the landslide polygon.

$$D_{ch} = \sum_i d_{ch,i} - min(d_{ch,i}) \quad and \quad D_{va} = \sum_i d_{va,i} - min(d_{va,i}) \tag{10}$$

The normalised channel proximity grid effectively provides a distribution of distances to the landslide toe (irrespective of curvature). While the normalised valley axis proximity distribution includes the effects of landslide toe curvature. Subtracting the median of the distance to toe distribution from that of the normalised distance to valley axis distribution then provides a description of the degree of landslide impingement into the valley.

$$D_L = med(D_{va}) - med(D_{ch}) \tag{11}$$

Importantly this calculation evaluates the area of the landslide toe rather than the more variable morphology that may be mapped in the toe region.

In order to compare frequency-amplitude distributions of planform channel deviations $D_F$ to those of the channel offset $D_L$, we use the landslide width as proxy for the half-wavelength $\lambda_{1/2}$ of the landslide. The landslide width is defined as the difference between the minimum and the maximum flow distance along the stream network within the landslide polygon. The flow distance being calculated along the stream network, the values for distance on the adjacent hillslope and landslides only increase parallel to the river channel and not perpendicular to it. The difference between the minimum and maximum values of

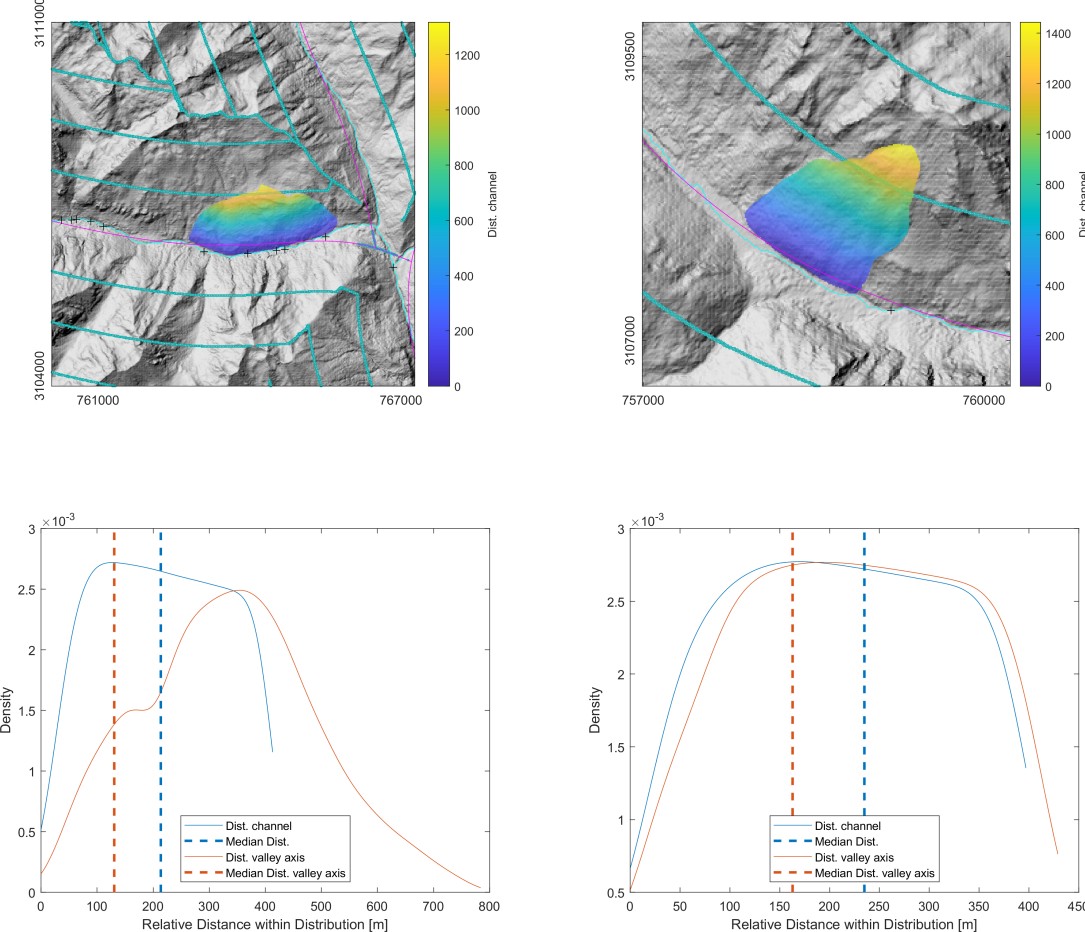

**Figure 6.** Examples of distance distributions for two landslides. Top: Distance away from the channel within the landslide polygon, Bottom: Kernel density estimates of the normalized distributions of distance away from the channel (blue) and distance within the valley axis grid (orange) and their respective medians (dashed). The difference between the two medians corresponds to the planform channel offset $D_L$. The landslide outlines are based on Dini et al. (2020), the hillshade was derived from (JAXA, 2017).





flow distance in the landslide polygon therefore roughly correspond to the lateral extent of the landslide in a reference system

parallel to the river channel.

$$\lambda_{1/2,i} = max(D_{us,i}) - min(D_{us,i}) \tag{12}$$

## 2.3 Evaluation of rates of channel and hillslope displacements

We can further investigate the interaction between landslides and river channels by evaluating rates of landslide displacement and river channel evolution over equivalent time periods. Distributed catchment-wide hillslope erosion data indicate recent

rates in this region are just less than 1 mm yr$^{-1}$ (Portenga et al., 2015). Assuming hillslope erosion keeps pace with channel incision, we require at least 10-20 kyr, for the long-term trend of the longitudinal river profile evolution to exceed the 10 m vertical resolution of our DEM at the valley axis and be differentiable from local noise in the DEM. Nonetheless, we can use morphological markers to evaluate long-term ($> 10^4$ kyr) channel and hillslope evolution. In particular, $\chi$-analyses of river profiles, provide a means of normalizing upstream distance by the upstream drainage area. By normalizing values of $\chi$ by

erodibility $K$, we can obtain $\tau$ which has units of time (Whipple and Tucker, 1999). $\tau$ corresponds to the amount of time that has passed since an erosional signal (like a knickpoint) has travelled in $\chi$-space from one point to another:

$$\tau = \frac{1}{K}\chi \tag{13}$$

Employing equation 5 we calculate $\chi$ for the Paro and the Punakha basins of western Bhutan, starting at the range front at 100 m a.s.l. using the TopoToolbox function *chiplot*. Inputs to this function are the stream network derived from the DEM, as

well as a concavity ($\theta$) value of 0.45, which is well supported in literature (Snyder et al., 2000; Duvall and Burbank, 2004) and yields linear profiles in the fluvially active regions of NW Bhutan.

$K$ is a collective term to capture the effects of (e.g.) rock mass strength, tectonic overprint, and climate or erodibility. These properties are difficult to assess individually, but a typical values of $K$ ($10^{-7}$-$10^{-5}$ m$^{0.1}$ yr$^{-1}$) are reported in literature for mountainous regions (Scherler et al., 2017; Leith et al., 2018). Considering the subtropical setting with high weathering rates

of our study area we use a value at the upper end of the range ($10^{-5}$ m$^{0.1}$ yr$^{-1}$) for our analysis.

We estimate the amount of time that has elapsed at the location of a landslide between the passing of a knickpoint (i.e. initiation of landslide formation) and the present-day by measuring the distance between landslides and knickpoints in terms of differences in $\chi$ at the two locations ($\Delta\chi$). For this purpose, we associate to each landslide a knickpoint, which is located upstream of the landslide toe. In the case of several knickpoints being found upstream of the landslide toe, the largest knickpoint

in terms of vertical offset ($\Delta z$) with respect to the idealized river profile is chosen. Landslides upstream of which no knickpoint was found were discarded for the further analysis.

We calculate rates of channel offset as a result of landslide displacement ($R$) by assuming that the present-day channel offset ($D_L$) corresponds to the total amount of channel offset caused by the landslide since the passing of the knickpoint. In order to





obtain a rate of displacement, we then divide the offset measured at the landslides by the $\Delta\tau$ of the landslide-knickpoint pair,
which corresponds to the value for normalized time $\Delta\chi$ by an estimate for $K$:

$$R = \frac{D_L}{\Delta\tau} = K\frac{D_L}{\Delta\chi} \tag{14}$$

We may expect that for a given landslide displacement rate, $R$ will decrease with increasing stream power. In terms of Eq. 6, increased stream power will lead to an increase in the velocity of particle impacts, and therefore the channel wall erosion rate at the landslide location. Given the constitutive properties of the landslide material are likely weaker than the opposing bank,
the increasing stream power will disproportionately increase the erosion rate of the landslide toe (with respect to the opposing bank), and tend to straighten the channel, or reduce the rate of channel offset.

## 3  Results

### 3.1  Characteristic channel sinuosity

The fractal nature of stream networks was first described by Tarboton et al. (1988), who demonstrated, in analogy to Mandelbrot
(Mandelbrot and Wheeler, 1983), that the total length of a stream network is function of the the the 'length of the ruler' with which it is measured and proportional to it to the power of $1 - D$, with $D$ the fractal dimension. However, the planform morphology of the stream network with respect to the 'length' and 'amplitude' of channel sinuosity has not been approached in this manner to the present day.

We find that the power spectra of the planform channel deviations (i.e., channel sinuosity) follow a power law over a range
of 3 orders of magnitude ($10^{-4}$-$10^{-1}$ m$^{-1}$, Fig. 7). The amplitudes range from $10^{-3}$-$10^1$ m for higher frequencies ($\sim 4\cdot10^{-2}$ m$^{-1}$) and from $10^1$-$2\cdot10^2$ m for lower frequencies ($\sim 2\cdot10^{-4}$ m$^{-1}$).

The power-law relationship we observe in our study between the half-wavelength and the amplitude of the planform deviation follows a power-law with an exponent of 1.1. This can be expressed as $1/f^a_{noise}$, where $a$ is the power law exponent. This behavior is known as pink noise and has been described for a range of natural systems, such as coastlines (Mandelbrot and
Wheeler, 1983; Baas, 2002), topography (Chase, 1992), and mixing of chemical solutes in water (Kirchner and Neal, 2013). Such systems are self-organizing, meaning that they develop toward a fractal equilibrium which is independent of the size of the original forcing, but instead governed by the scale of the smallest (length-)unit within the system.

### 3.2  $\chi$ analysis and knickpoint distribution

We identified 51 knickpoints in the study area, with most of them clustering at the upstream end of high excess topography
regions (Fig. 2). Only a few knickpoints are found in the Southern Valleys with all of them clustering at the upper limit of the geomorphic domain, just below the Inner Valleys where in turn only one knickpoint was identified. The low erosional activity of this region is also reflected by the broad alluvial plains, shallow hillslopes, low excess topography (mean: 16.5 m in Paro, 28.8 m in Punakha) and low $k_{sn}$ (mean: 5.1 m$^{0.9}$ in Paro, 6.9 m$^{0.9}$ in Punakha). The largest concentration of identified knickpoints





Earth **Surface**
**Dynamics**
Discussions

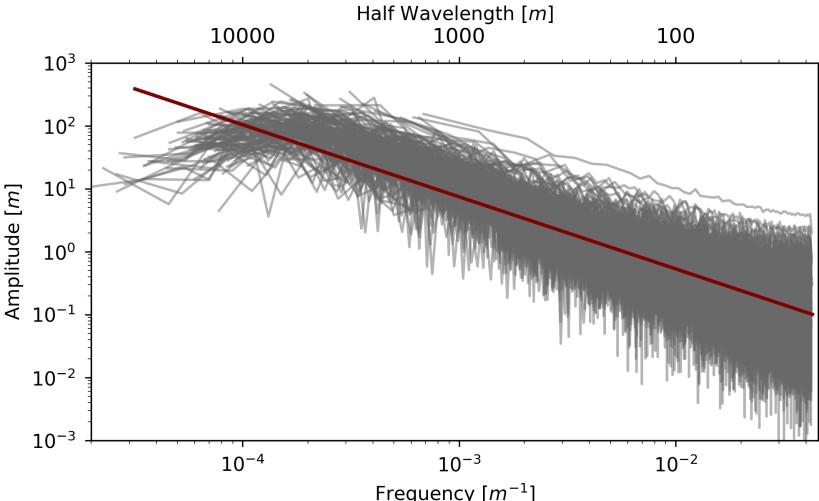

**Figure 7.** Power spectra of the planform channel deviations. Each line corresponds to the power spectrum of one channel reach.

is located in the domain upstream of the Inner Valleys, at the upper end of the Northern Valleys. Here, the erosional activity is

higher with respect to the Inner Valleys, which is expressed in the deeply incised fluvial valleys, steep hillslopes, high excess

topography (mean: 45.5 m in Paro, 80.8 m in Punakha) and high $k_{sn}$ (mean: 8.1 m$^{0.9}$ in Paro, 13.5 m$^{0.9}$ in Punakha). Several

knickpoints are also found in the glacially overprinted landscape of the High Himalaya. In this domain fluvial erosional activity

is again reduced and values for both $k_{sn}$ and excess topography are low (mean $k_{sn}$: 14.7 m$^{0.9}$ in Paro, 10.7 m$^{0.9}$ in Punakha

and mean excess topography: 26.2 m in Paro, 49.2 m in Punakha respectively)

Between the two major basins of the study area, stark differences in the vertical location of the profiles are found in the

zones of the southern gorges and those of the Inner Valleys, where the western basin (Paro) is located 1000 m higher than a

geomorphologically similar zone of the eastern basin (Punakha). Above this zone however, the two profiles overlap, showing

very similar shapes in both the Northern Valleys and the glacially dominated High Himalayan Plateau, suggesting similar

erosional and tectonic conditions in both basins during a period prior to the formation of the alluvial plains.

**3.3 Planform channel offset**

In order to evaluate whether the flow path offsets attributed to landsliding exceed those typical of the stream network, we

compare the measured offsets to the amplitude of channel deviation at an identical $\lambda_{1/2}$ derived with the Fourier transform

(Fig. 8). 477 of the mapped landslides in the study area are located in close vicinity to the stream network, i.e. at a distance

from the channel that less than a third than their length and were were included in the calculation of the channel offset. The

values cover a range from -62 to 883 m, with two thirds of the positive offset values smaller than 64 m. In order to semi-

quantitatively assess our offset metric, we measured the offset of 100 landslides by hand. Results of this analysis are presented

in Appendix C1 and indicate a linear correspondence with our automated method, which typically over estimates displacements





by 17% (mean calculated offset: 68.5 m, mean measured offset: 82.5 m). The channel offset of about 83% of the landslides exceeds the average amplitude of deviation which would be expected based on the stream network at a given wavelength.

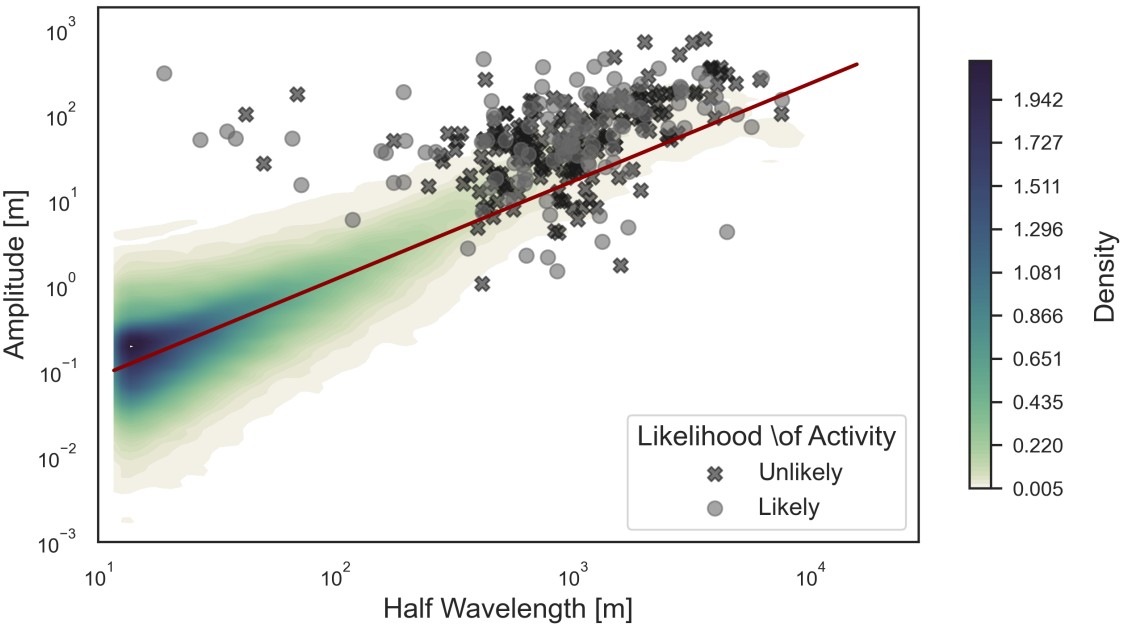

**Figure 8.** Spectra of the planform channel geometry (shades of blue) and the planform channel deviation of the landslides. Markers represent the likelihood of the mapped feature being an active landslides (Dini et al. 2019)). On the x-axis $1/f$ is shown, corresponding to either half-wavelength of the planform channel deviation $D_F$ or width of the landslide. The y-axis shows amplitude corresponding to $D_F$ or the channel offset measured in the landslides ($D_L$) respectively.

## 3.4 Relative rates of channel and hillslope displacements

In order to calculate relative rates of hillsope and channel displacement, we first have to associate each landslide with an appropriate knickpoint. Knickpoints were associated to roughly a third of the landslides within the elevation range of the landslide (135 cases), while in 265 cases knickpoints further upstream than the elevation range were chosen. In 512 cases no knickpoint was associated. This was usually the case at the upper end of the channel profile, where the river profile transitioned into a glacial morphology or the channel head was reached. For 226 of the landslides it was possible to determine both a channel offset and an associated knickpoint. For these landslides the channel offset rate $R$ was derived with 75% of the values ranging between $9.1 \cdot 10^{-3}$ and $7.6 \cdot 10^{-2}$ m m$^{-1}$. The units of this 'normalized' rate are [m m$^{-1}$] since the channel offset is divided by $\chi$, i.e. normalized time, which has units of [m].



Earth **Surface**
**Dynamics**
Discussions

The relationship between the channel offset rate $R$ – derived from the channel offset measured in the landslides and the

distance between the landslide and its associated knickpoint in $\chi$-space – and the local stream power based on the normalized

steepness index $k_{sn}$ follows a power-law relationship, with increasing displacement rate for increasing $k_{sn}$ (Fig. 9). This

suggests that locations with higher $k_{sn}$ are subject to higher channel offset rates. No clear trend with respect to the landslide

size can be made although the largest landslides (3-10 km$^2$) tend to be associated with both higher channel steepness indices

and relative offset rate.

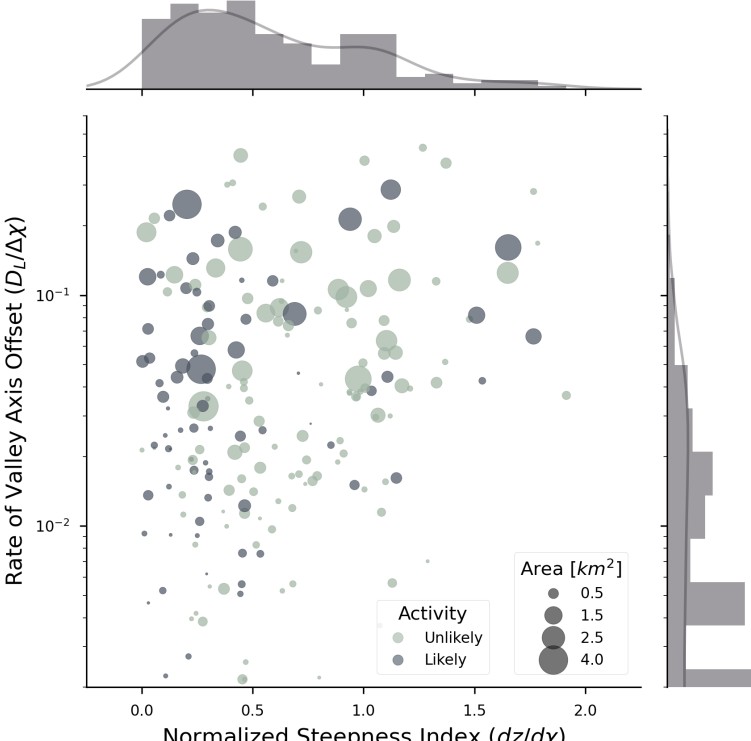

**Figure 9.** Normalized steepness index ($k_{sn} = dz/d\chi$) vs. rate of channel offset $R$ measured in the landslides. The markers are scaled by the landslide surface area and colored by their assigned likelihood of activity.

## 4   Discussion

### 4.1   Characteristics of geomorphic domains

Our combination of river profile analyses, knickpoint identification, excess topography analysis provide new insight into the

interaction of rivers and landslides across three geomorphic domains in the High Himalaya of Bhutan: the Inner Valleys (IV),



the Northern Valleys (NV), and the High Himalayan Plateau (HHP). The former two domains are fluvially dominated and
we observe increased interaction with the stream network, while the latter domain contains a mixture of fluvial and glacial
features. Correlation of fluvial and hillslope activity in these regions is therefore less clear, as we do not explicitly consider
morphological characteristics resulting from glacial and post-glacial activity in this study.

The prevalence of landslides is commonly assumed to be an indicator of hillslope activity, while knickpoint occurrence
similarly suggests enhanced erosional activity. We calculate landslide density by multiplying the number of landslides by total
landslide area divided by the area of the geomorphic domain, while for the knickpoints density is calculated by dividing the
number of knickpoints in a geomorphic domain by the area of the geomorphic domain. We find landslide density is inversely
correlated with that of knickpoints in each of the two fluvially dominated domains, where the IV contain a high density of
landslides and relatively few knickpoints (Fig. 10 (a) and (b)). The landslide density, on the other hand, is lower in the NV and
the knickpoint density is higher. The difference in landslide density between the two domains is stronger in the the Paro basin
(23 landslides [$m^2$ $m^{-2}$] in IV and 5 landslides [$m^2$ $m^{-2}$] in NV) and less pronounced in the Punakha basin (16 landslides
[$m^2$ $m^{-2}$] in IV and 15 landslides [$m^2$ $m^{-2}$] in NV). With respect to size the landslides are very evenly distributed with mean
landslide sizes in IV of 0.26 $km^2$ in Paro and 0.27 $km^2$ in Punakha and in NV with 0.24 $km^2$ in Paro and 0.26 $km^2$ in Punakha,
Fig. 10 (c)). The coexsitence of high landslide density and low knickpoint density (0 and 1 knickpoints in the Paro and the
Punakha valley respectively) suggests a disconnect between hillslopes and rivers in the IV and implies landslides were either
initiated by a previous fluvial regime and are now relict or the reduction in slope stability is related to additional factors (e.g.
weathering).

Further insight into the connectivity between landslides and rivers in the AV may be gained by integrating properties, which
capture the erosional potential such as excess topography and channel steepness. High values of both excess topography and
$k_{sn}$ are evident in the NV suggesting high erosional activity and high erosional potential in this domain, while both excess
topography and $k_{sn}$ are lowest in the IV (Fig. 10 (e) and (f)). This indicates that active and also future erosion are preferentially
occurring in the NV and that little erosional activity is currently taking place in the AV, supporting an interpretation that
landslides in the AV are likely relict or dormant features.

Finally, we can evaluate the connectivity between landslides and channels by assessing the proximity of landslides to chan-
nels. Landslides throughout the study region tend to be located close (within 113 m in the Paro Basin, and 43 m in the Punakha
basin) to the flood plain (Fig. 10 (d)). Generally, we see landslides in the NV are located closest to the channels (on average
188 m in Paro and 208 m in Punakha), while the IV demonstrate a looser clustering (on average 357 m in the Paro and 303 m
in the Punakha valley). The Punakha basin High Himalayan Plateau demonstrates a relatively weak association of landslides
with channels with an average distance of 419 m, likely reflecting generally steeper topography within 1 km of the valley axis
in the glacial valleys. The Paro basin HHP contains only three landslides, which are located adjacent to the NV-HHP boundary,
and the distribution relative to the channel does not reflect characteristic processes within the domain. The fact that landslides
are closely associated with the rivers in the NV supports an interpretation of stronger landslide - river interactions in this re-
gion, since channels in direct contact with the landslide can erode landslide material from its toe and in turn be supplied with
sediment by the landslide.



Earth **Surface**
**Dynamics**
Discussions

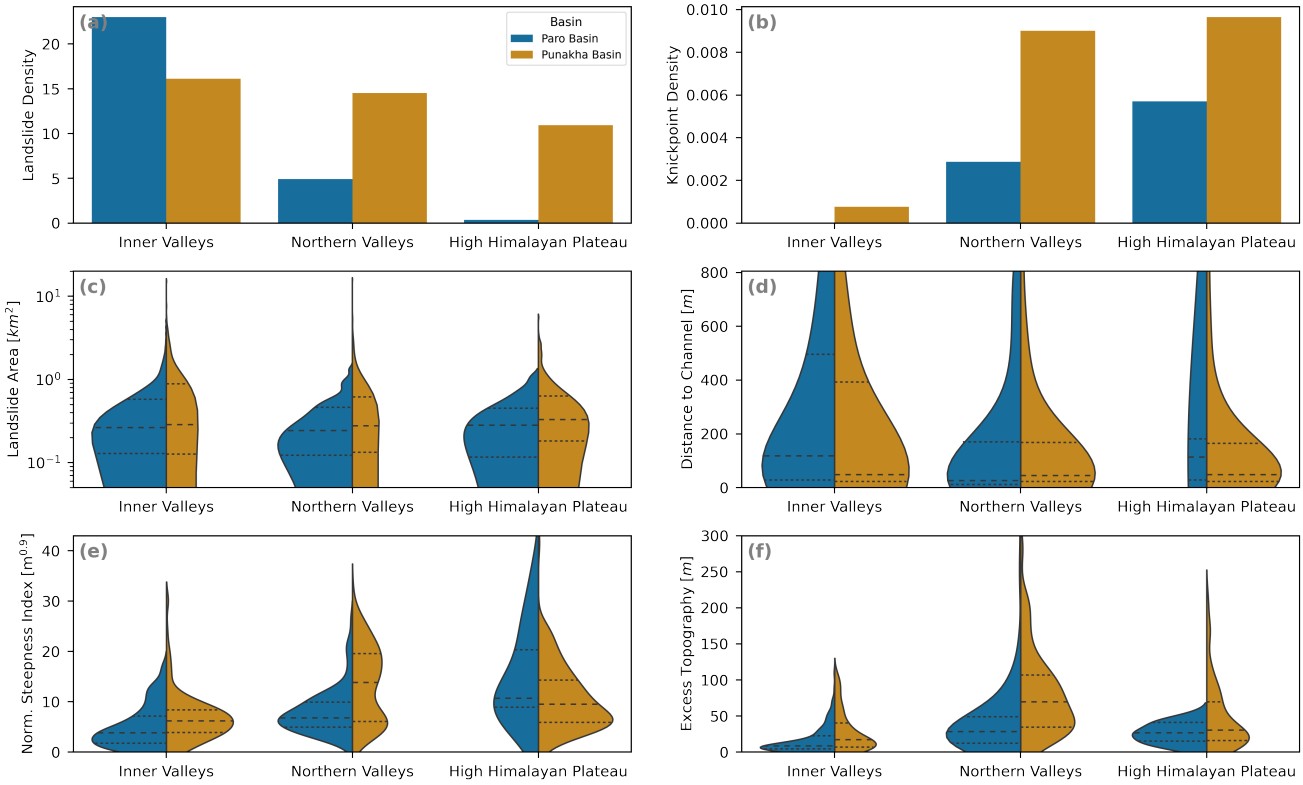

**Figure 10.** Distributions for each geomorphic domain and basin of key variables. (a) landslide density (number of landslides multiplied by total landslide area and normalized by the area of the geomorphic domain), (b) knickpoint density (number of knickpoint per area of the geomorphic domain), (c) landslide area, (d) distance of the landslide to the nearest channel, (e) normalized steepness index $k_{sn}$, and (f) excess topography. Dashed lines correspond to the mean of the distribution and the dotted lines to the quartiles (25% and 75%).

## 4.2 Landslide activity

The apparent tendency toward heightened erosional potential, and increased coupling between hillslopes and rivers in the NV can be further investigated by identifying planform channel offsets, and (assuming landslide activity was initiated by the passing of a migrating knickpoint) relative rates of vertical and lateral channel offset. In order to analyse only those landslides interacting with the stream network at the present day, we consider only those which are within 50 m of the present day river channel alignment. The channel offset distributions are centered above 0 in all domains (Fig. 11 (a)), indicating that most

landslides ($> 83\%$) have offset the channel at their toe. The range over which the distributions are spread is larger in the Inner Valleys (standard deviation 99 and 191 m for the Paro and the Punakha basins respectively) than in the Northern Valleys (standard deviation 33 and 49 m for the Paro and the Punakha basins respectively). This supports the notion that the Northern





Valleys are a geomorphologically younger domain than the Inner Valleys, where topography has had more time to react to passing erosional signals and consequently has had more time to form large landslides, which displace the channel.

In order to be account for differences in stream power along the stream network when evaluating channel displacement rates we normalise the rate of channel offset by the normalised steepness index $k_{sn}$ representing stream power.

$$R_n = \frac{D_L/\Delta\chi}{k_{sn}} = \frac{D_L/\Delta\chi}{dz/d\chi} \tag{15}$$

The normalized rates of channel offset show half an order of magnitude differences between the Inner Valleys and the Northern Valleys (Fig. 11 (b)): in the Northern Valleys 75% of the landslides have normalized rates lower than 0.1 $m^{0.1}$ $yr^{-1}$ in

the Paro valley and 0.06 $m^{0.1}$ $yr^{-1}$ in the Punakha valley. In the Inner Valleys 75% is reached at normalized rates of 0.38 $m^{0.1}$ $yr^{-1}$ in the Paro valley and 0.23 $m^{0.1}$ $yr^{-1}$ in the Punakha valley. We suggest that additional time available for progressive failure of the rock slope and formation of the slope instability in the IV leads to higher rates of displacement.

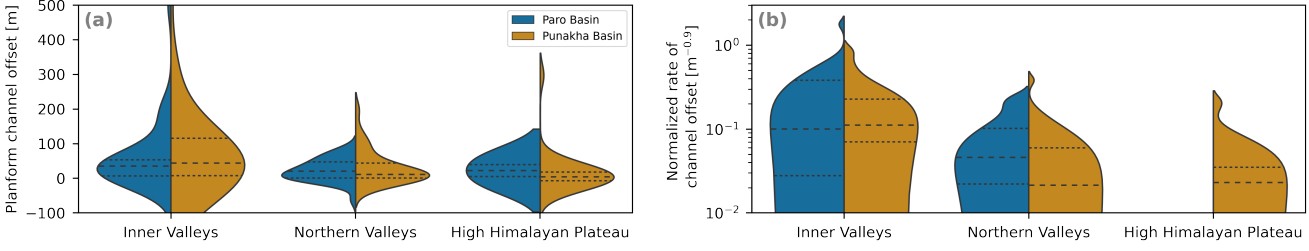

**Figure 11.** Distributions of the channel offset $D_L$ (a) and the normalized rate of channel offset $R_n$ (b) for each geomorphic domain and basin for landslides less than 50 m away from the river channel. Dashed lines correspond to the mean of the distribution and the dotted lines to the quartiles (25% and 75%). The distributions are normalized by their width.

The spatial distribution of landslides supports this observation, as landslides tend to be located downstream of a knickpoint and the corresponding increase in excess topography (Fig. 12). Landslide dimensions do not seem to correlate with normalized

rates of channel offset, as we observe a wide range of sizes, which do not seem to follow a spatial pattern.

Negative values of channel offset are due to a methodological artefact, but nonetheless carry information regarding the interaction of the landslide with the stream network. In equations 10 and 11 we see that negative values for $D_L$ are obtained if the median value of the distance distribution within the landslide polygon to the valley axis is larger than that to the channel. This may be the case when the shape of the polygon protrudes less with respect to the valley axis than with respect to the river

channel, which can for example occur when material from the landslide toe is eroded more efficiently than can be refurnished by the landslide offset. A further caveat needs to be mentioned with respect to the channel offset measure $D_L$ itself, which is treated as a proxy for the sediment input of the landslides to the stream network in this study. It is however likely that both sides of the valley contribute sediment to the river channel, yet at different rates. The measured offset is therefore a proxy for *relative* sediment input of each valley side.



**Figure 12.** Normalized landslide-induced channel displacement: map of a zoom on the Punakha valley with knickpoints, excess topography, and the landslide inventory colored by the normalized rate of channel offset $R_n$. Grey landslide polygons correspond to landslides for which no value of $R_n$ could be calculated. Excess topography and the hillshade were calculated based on two DEMs (JAXA, 2017; NASA JPL, 2013) and the landslide outlines are based on Dini et al. (2020).





We can use our method to analyse the temporal evolution of large creeping landslides which is likely to reflect a non-linear evolution with time. The temporal evolution of channel offset due to landsliding as a function of (normalized) time is shown in figure 13. In order to account for stream power, we normalize the channel offset rate $R$ by $dz/d\chi$ ($k_{sn}$) and obtain the normalized rate of channel offset $R_n$. With $\Delta\chi$, the distance in $\chi$-space between the location of the landslide and its associated knickpoint, as a proxy for the normalized time since the knickpoint has passed on the x-axis and with a rolling average (window

$= 100$ data points) over all of the values, we can retrace the above-mentioned stages of slope failure. We observe a decrease in $R_n$ for values of $\Delta\chi$ up to $\sim$1400, which corresponds to the initial phase of surficial adjustment to the new topography. Beyond this point $R_n$ increases steadily up to values of $\Delta\chi$ of $\sim$3000, which we interpret as a phase where the accumulated damage in the rock slope is progressively expressed in landsliding. Only few data points are available at values of $\Delta\chi$ higher than $\sim$3000 and no definite statement can be made here. While reducing driving stresses (for example through removal of driving mass

or toe buttressing) may be an effective means of stabilising maturing landslides, younger landslides ($500 \geq \Delta\chi \geq 1500$) are likely to be especially sensitive to (for example) changes in pore pressure and other factors reducing stability. Mature landslides ($\Delta\chi \geq 1500$) are likely highly degraded and are likely to continue to creep irrespective of changes in boundary conditions.

This plot provides unique new insight into the temporal evolution of large creeping landslides in mountainous regions. Once a wave of increased erosion has passed the location of a rock slope, the part of the slope, which is unstable in the new geometry

will fail in smaller landslides. With time most of the immediate instabilities will have failed leading to an intermediate pseudo-stable period, during which the rock slope does not predominantly produce apparent failure, but rather will the rock mass strength be progressively reduced due to the increased differential stresses. Once the accumulated damage in the rock slope has produced a (more or less continuous) failure plain, a second wave of rock slope failure initiates, potentially with larger landslide volumes.

Our new method provides important new tools to evaluate and perhaps quantify the long-term activity of mapped landslides. This can potentially add to future landslide hazard assessment studies in the context of hydro power dam construction as our results provide both an indicator of mean long-term activity and the evolutionary stage (and therefore stability state) of large creeping rock slope failures. In addition, quantifying landslide activity can serve as base for estimating sediment input into the dam reservoir.

**5    Conclusions**

In this study we identify potentially creeping landslides in NW Bhutan by comparing profile and planform morphology of the stream network to apparent channel displacements derived from a mapped landslide inventory. We show that we can quantify the planform river channel offset caused by landsliding by extracting distance distributions within the landslide (polygon) and comparing to characteristic channel sinuosity at similar wavelengths.

By applying the Fourier transform to metrics of the stream network (planform deviation from the valley axis), we obtain a fractal distribution of the amplitude and wavelength of the river channel sinuosity and show that it follows the power law of pink noise ($1/f_{noise}^a$) with an exponent of $a = 1.1$. When comparing the power law of the planform channel deviation to that of



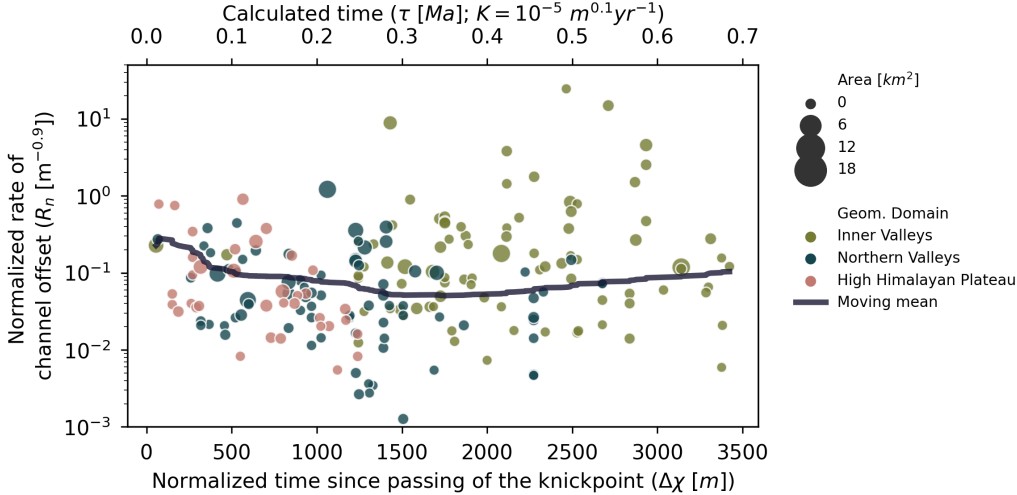

**Figure 13.** Normalized rate of channel offset $R_n$ as function of time. Bottom x-axis: normalised time $\Delta\chi$, top x-axis: calculated time $\tau$ in million years ($K = 10^{-6}$ m$^{0.1}$ yr$^{-1}$, $n = 1$, $m = 0.45$). The size of the markers is scaled to the landslide area. Color represents the geomorphic domain. The dark grey markers correspond to the moving mean over the data (window of 100 data points). The hillshade was derived from (JAXA, 2017) and the landslide outlines are based on Dini et al. (2020).

the channel offset measured in landslides with respect to their width, we find a good overlap between the two distributions with a clear trend of higher amplitudes at a given wavelength in the landslide channel offsets with respect to the planform channel deviations. This supports the hypothesis of a close interaction between the stream network evolution, hillslope development, and particularly hillslope erosion through landsliding. We make a case for increased landslide formation in geomorphic domains demonstrating high channel-hillslope connectivity with highest average landslide activity reached in formerly well-connected domains.

We calculate normalized landslide-induced channel displacement rates and provide new insight into the temporal evolution of landslides in this setting. We identify two major phases in normalized displacement rates with respect to passing erosional waves: an exponential decrease – likely reflecting a reduction in driving force as valley walls relax to a more stable profile – until a calculated time of 300 ka (mostly occurring in the well-connected geomorphic domain) and followed by an increase in activity, that may reflect a dominance of rock mass degradation (mostly occurring in the less well-connected geomorphic domain).

Our new method provides important new tools to evaluate and perhaps quantify the long-term activity of mapped landslides. While the methods of planform stream network analysis and channel offset measurements need to be tested in different settings, we believe that the integration of geomorphological and long-term landscape evolutional concepts and methods in the analysis and characterization of creeping landslides is a valuable addition to hazard assessments for hydro power dam constructions in mountainous environments.



*Data availability.*   The landslide polygons are based on the landslide inventory by Dini et al. (2020) available at 10.3929/ethz-b-000387601.


Earth **Surface**
**Dynamics**
Discussions

## Appendix A: Knick point tolerance (sensitivity analysis)

**Figure A1.** Sensitivity analysis for a range of the tolerance used in the identification of knickpoints. An increase in number of identified knickpoints is observed at a tolerance value of 100.





## Appendix B: Normalized steepness index in tributary channels

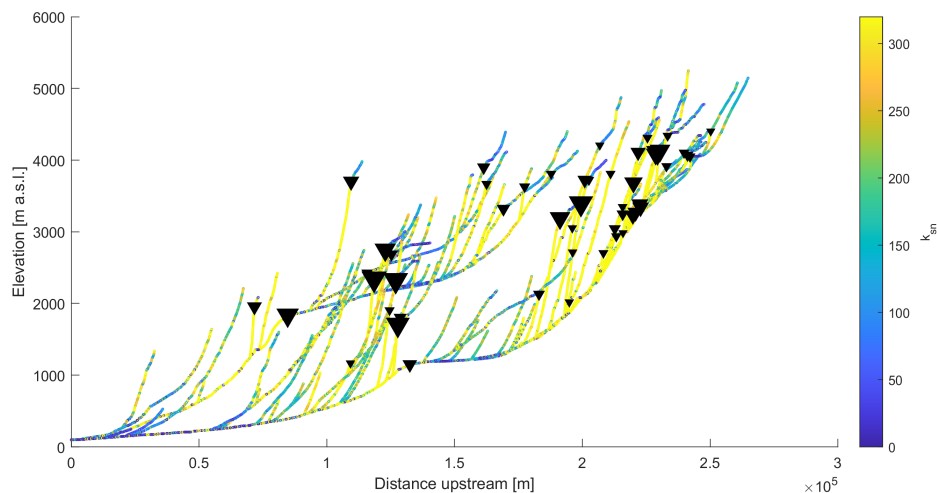

**Figure B1.** River profiles of the two major basins with the channels colored by $k_{sn}$ and the knickpoints scaled by vertical offset between the idealized and the original river profile.





# Appendix C: Planform channel offset

## C1 Measured vs. calculated planform channel offset

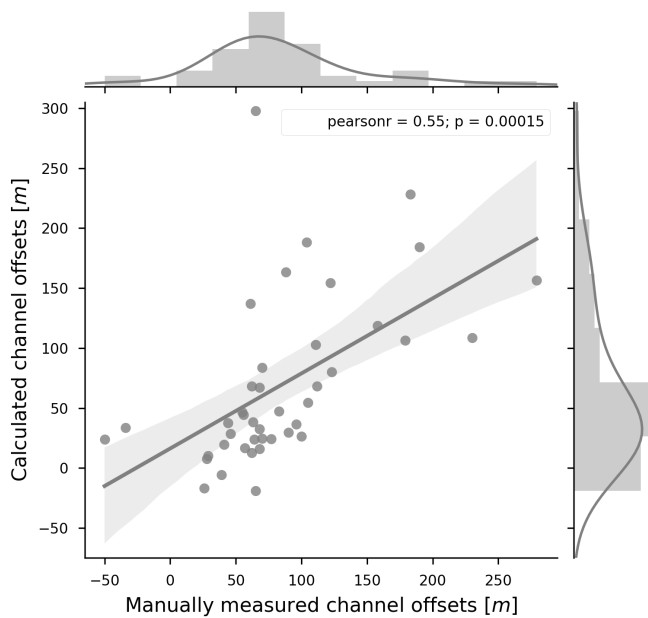

**Figure C1.** Measured vs. calculated channel offset for a subset of the landslides.

## C2 Planform channel offset vs. normalized time





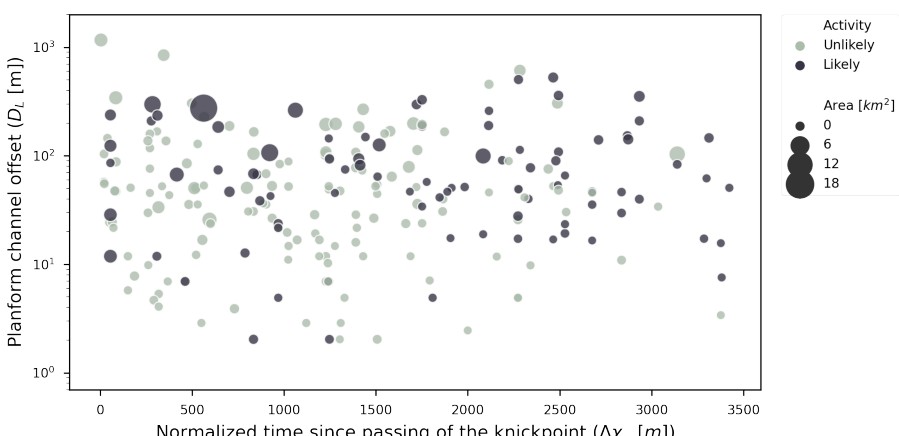

**Figure C2.** Planform channel offset $D_L$ as function of normalised time $\Delta\chi$. The size of the markers is scaled to the landslide area. Color represents the likelihood of activity.



*Author contributions.* Conceptualization and methodology were developed by LP and KL. Field investigations were carried out by all authors. Project administration and supervision were performed by SL and KL. Visualization and writing of the initial draft was done by LP. The initial draft was revised by KL and commented by SL.

*Competing interests.* The authors declare no competing interests.

*Acknowledgements.* The authors would like to thank Jigme Thinley and Dr. Phub Dorji from the College of Natural Resources of the Royal University of Bhutan and Walo Bertschinger AG for their support and advice during field campaigns. We are grateful to Dr. Helen Beeson, Dr. Odin Marc, and in particular Dr. Eric Deal for their valuable input and discussions regarding fluvial geomorphology. Many thanks go to Dr. Peter Heinzmann and Nicolas de Palézieux who gave valuable insight to the application of the Fourier transform to planform river morphology. This work was funded by an ETH grant (ETH-38 15-2) and an ESA Alcantara project (ESA 4000117652/16/F/MOS).





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
