# Peer review of "Planform river channel perturbations resulting from active landsliding in the High Himalaya of Bhutan"

_Earth Surface Dynamics, 2020_

## Referee Comment (RC1) · Anonymous Referee #1 · 2 Jan 2021

Comment to Authors Title: Planform river channel pertubations resulting from active landsliding in the High Himalaya of Bhutan Journal: Earth Surface Dynamics Corresponding author: Larissa de Palézieux Co-authors: Kerry Leith, Simon Loew

General remarks The manuscript presented by Palézieux et al. deals with the quantification of lateral channel migration induced by large creeping landslides. They used Fourier transform to separate natural channel amplitudes from landslides-produced variations. Using the inventory of Dini et al. (2020), they show that a large majority of creeping landslides cause lateral channel migration. Furthermore, they hypothesize that creeping landslides are primarily triggered by migrating knickpoints. Under

this assumptions they back-calculate the timing of the onset of creeping landslides showing that a phase of activity alternated with a phase of slower movement/inactivity. Lateral channel migration produced by creeping landslides is not very well quantified and the usage of Fourier transform a somewhat unique approach. Combined with the approach to constraint rates of activity the manuscript shows enough originality to be published. However, the authors should modify, clarify and/or reassess a couple of points. The manuscript contains a few assumptions which have to be discussed more in detail. The authors mention that seismic activity in the study area is low. This observation is explained in the introduction. Hence, seismic activity is not responsible for triggering creeping landslides. So the authors assume now that knickpoint migration is mainly responsible for triggering creeping landslides. But what about hydrological and climatic influences? What about lithological weak point? Furthermore, migrating knickpoints are primary produced by seismic activity. How do the authors explain the knickpoints in the first place? Especially influences of climate as well as lithology have to be discussed more in detail. Another point are the methods, which seems to contain a few flaws. Especially when calculating the planform channel offset DL. The calculated channel axis, probably derived from the DEM, is crossing the hillslope (Figure 6). Resulting in negative values in the distance distribution of the landslides. Especially the distance distributions (Figure 6 bottom) pose some questions. Please, check the comments in added in the supplement file. Especially the discussion needs further improvement since a lot of the written paragraphs belong to the Methods or Results. Rarely any of the Results are put into perspective with previous research. Regarding the general structure: The introduction is too long (10 pages). Therefore, I would recommend to revise how necessary certain explanations of methods are (e.g. lateral channel migration is explained in the Introduction as well as in the Methods) and if there is the possibility to add another section "Regional setting". Regarding the sentence structure and usage of scientific terminology: Even though methods are explained in quite the detail, often terminology is used without proper definition and in varying contexts. Examples are mentioned in the technical details. Furthermore,

sentence structures are often too long and too complicated which results in grammar mistakes, sometimes leading to sentences which are hard to understand.

Please also note the supplement to this comment:
https://esurf.copernicus.org/preprints/esurf-2020-85/esurf-2020-85-RC1-supplement.pdf

**Supplement:**

**Specific comments to Authors**

Title: Planform river channel pertubations resulting from active landsliding in the High Himalaya of Bhutan
Journal: Earth Surface Dynamics
Corresponding author: Larissa de Palézieux
Co-authors: Kerry Leith, Simon Loew

Abstract

L1: What do you mean with persistent features? They are numerous and abundant? Or one landslide stays active a long time?

L1: Evolution is meant to be stability?

L2: Why is it difficult to assess hazard? Features are difficult to detect? They occur in inaccessible areas? Maybe add a half-sentence about the reasons.

L3 ff: Rather explain the reasons for using Fourier transform first and then succeeding explain the results

L3/4: Why is it important to mention that the wavelength is consistent with a fractal distribution (pink noise with a specific exponent)? How does it help us to distinguish areas with creeping landslide?

L4/5: Maybe it would help the reader to first mention that river wavelength is disturbed by landslides.

L7: Exceptional amplitudes of what? Landslide activity? Or river movements?

L7: What other geomorphological processes can induce lateral channel migration as well or if it is just creeping landslides?

L8/9/10: This sentence is hard to read. Maybe splitting the sentence? Or switch "range" to "ranging"

L10: What do you mean with "early period"? How early is early and how long is a long period? Quantify the words with numbers.

L10: Are the rates an interpretation of your normalized channel offsets? If so are they really "consistent" or are they rather supporting the interpretation of phases of different landslide activity?

Introduction

L19: Why only "parasitic" failures? And not just slope failures in general?

L20: What would be sensitive mountain infrastructure? Isn´t all infrastructure human-made by definition?

L21: What do you mean with "lessen dam stability requirements"?

L21/22: What are other "driving geomorphological changes" that can affect dams than landslides? Either define it as landslides or mention examples.

L23: Maybe not just construction workers? Do not all humans depend on the functionality of the dam?

L25: Channel displacement due to landslide activity is quite well known (e.g. Korup, 2005, Earth Surf. Process. Landforms 30, 783– 800). However, it seems that your goal is rather to investigate whether channel displacement can be specifically linked to creeping landslides and how the channel is disturbed and adapts during longer time-spans. Be more precise in your usage of "landslides" and "creeping landslides" and try to specify your goals in these section.

Z25: Do you really "identify" zones of heightened erosional activity (creeping landslides?)? Or do you rather check whether landslides activity is reflected by lateral channel migration? You cannot do both at the same time.

L26: Evolutionary stage = Landslide activity?

L28: What are examples of modification of bedrock channels? What do you mean exactly?

L29: What do you mean with "those"? Fluvial processes? Uplift and erosional processes? It´s ambiguous here. And so you state that erosional processes (in general) are not influencing the hillslopes? However one sentence after it you mention landslide as erosional features.

L30: Are there other "Hillslope processes" that affect both bedrock strength and sediment transport? If not, leave out "such as".

L34: Evolution = migration?

L35: What do you mean with "cluster term"?

L35-L40: In this section it becomes clearer that you use specific terms without clarifying their meaning in the beginning. What is hillslope erosion (creeping landslides, landslides, slope failures, dry ravel?). What is erosional efficiency? Try to make the section easier to read and to follow your thoughts. So, erodibility depends on bedrock strength and erosional efficiency (also stream power?)? (Why use back-calculate?). State that you want to incorporate information about creeping landslides to study the lateral migration of channels. Again what do you specifically mean with "river evolution"? What do you focus on? Be clearer! Avoid using "evolution" so much, as it is a quite general term meaning nothing, everything, and all in between. What does it mean to "contextualize […] hillslope processes"? This sounds much like an empty phrase, specifically as you mention landslides right afterwards.

What do you mean with "engineering geological framework? You mentioned river dams in the beginning, maybe include as an example closing the circle?

L41-44: This section is so much easier to read! Use it and cooperate it into the section beforehand.

L47: Characteristic for… mountains? Plains? Alluvial fans? Be more precise!

L48: Again… evolution. Try to define it. What is your understanding of landscape evolution? Over what time frame? You mention relief production in the next sentence. Maybe it´s that?

L49: Avoid mentioning of the same word in one sentence ("limited"). Suggestions: Controlled

L51: Try to keep it simple "Sediment evacuation from active hillslopes" = erosion

L51: How does sediment comminution affect spatial and temporal erosion patterns? If you state something like this you should definitely put in a reference and explain further 1) the principle behind 2) Why is it important for your work?

L52: Help the reader to understand your work by mentioning the most important principles. What are erosion products? How does it influence tectonic orogens specifically? Concentrate on the

principles important for your hypothesis. If it is not important, delete it, otherwise explain why it is helpful to understand the kind of erosional product.

L56: "Drainage area is therefore regularly used as a proxy for river discharge". You referenced Wobus et al. (2006). However he only mentioned the typical power-law function that slope depends on drainage area (what you also mention later).

$$S = k_sA^{-\theta},$$

S represents local channel slope, A is the upstream drainage area, and ks and θ are referred to as the steepness and concavity indices.

Please, check again the sources or review the statement.

L56: What are *exponential* profiles? You mean longitudinal profile or river profile? Or you mean exponentially increasing slope?

L62: How long are "timescales relevant to relief production in alpine regions"?

L82 ff: Here you could try to keep it simpler as well. State that knickpoints lower base-level, increase local relief and reducing stability of adjacent slopes increasing the probability of failures.

However these failures would seem to be catastrophic. So we are probably not talking specifically about creeping landslides?

L86: Landslides are always driven by gravitation. However triggers usually include high and long rainfall duration as well as seismic activity (and anthropogenic triggers like road cuts). Be careful and distinguish between drivers, triggers and causes!

L87: I do not understand what you mean with "External support" when mentioning resisting forces. Resisting forces are usually a function of shear strength aka. cohesion and internal friction and depends on the rock type.

L88/89: Fluvial incision (in my experience) just affects the driving forces. It would interest me how resisting forces are changed (see comment above that resisting force is depend on rock type). Even when you mention that support at the slope toe is removed this affects only driving forces as it does not change material properties.

L90: Why is it a relative increase? Relative to what?

L90: How much is markedly?

L91: What do you mean with damage propagation? Joints, fissure and joints? Or just activity?

L92: How do you define compound landslides? What are with other types of slope failures?

93: The example you mention for increased landslide activity has a complete other setting and the reason for increased landslide activity is rather due to fluvio-glacial valley deepening and seismic activity than due to knickpoint propagation (Prager et al. (2008) did not mention knickpoints once in the paper). I would highly recommend to pick an example suited for your setting.

L94 ff: Why is it important for your work to calculate excess topography? You mention the method without explaining the advantage for your specific study.

L114 ff: Why not put the whole section in the Methods?

L118: Are there any other processes disturbing channel walls?

L148: The last part of the sentence already sound like a discussion/conclusion. This should be written there and not in the Introduction.

L151: What do you mean with "the manner"? Timing, velocity, phases of activity of the creeping landslide?

L151: What do you mean with evolution? Rather morphology, right?

L153: How many hydro-power dams are already built how many more are planned in your study area?

L155: What are geomorphic domains? River vs. Hillslopes? Either add the types or remove it.

L157: I would argue that there is no such thing as a "convergent tectonic setting". However, one of the three plate-tectonic settings is "convergent boundaries". Keep those definitions distinguished. The boundary would then be E-W striking.

L165: Magnitude 8 is a very big earthquake! Please check if there is really no seismic activity in the area. If so, that´s fine.

L172: Mention shortly what the five domains are before discussing them in detail.

L189: Every precipitation value should including a unit with year (e.g. 3500 mm/yr)

L200: What is the definition of "medium active" how many centimeters do the landslide have to move per year?

L200: Please revise whether the categories are either "Likelihood of activity" (as stated in figure 4 in the legend) or the "likelihood to be classified correctly as landslides". There is a huge difference between the statement that the landslide is "probably a landslide" or if the landslide is "probable active".

L206: You mean that the scar or the toe is maximal 0.5 km away?

L208: What does it tell us about the age of the landslides if there are no deposits?

Methods

L211: What do you consider as apparent? How much displacement would it be? Same question goes for "atypical". What do you consider "atypical"?

L214: What are "strong" variations? Until what number or percentage (or any other metric) do you consider any variations "as strong".

L218: What is your definition of a "flow network evolution"? How does it differ from increases (or changes in general) in sinuosity? If it is the same take one term.

L222-224: Why did you set those parameters as they are? 2.8 km² seems a rather arbitrary number.

L224: Why didn´t you use the just the 5 m ALOS?

L228: Why did you choose exactly 500 as stiffness? What happen when you try other values?

L234: What is the advancement of only using half-wavelength? Does it affect results of full wavelength are used? You could explain this further in detail.

L245: "Minimum value" of what?

L246: Name what $D_{ch}$ and $D_{va}$ since you do not mention the abbreviations beforehand.

L243 ff: In line 234 you mention that you focus just on the lower 1/3 of the landslide. However in figure 6 the whole landslides are marked and their distance from the channel axis and river are calculated for every pixel. This needs to be clarified.

When you calculate a distance it has to have a reference point. Do you calculate the distances within the landslide to a certain number of points on the river and the valley axis or just one?

Concerning figure 6: I also noticed that the red line (apparently the channel axis) is crossing the hillslope on both figures. This might show that the results are flawed. The channel axis has to be in the middle of the channel rather than on top of the slope. This might explain why the distributions are going into a negative area towards the left.

The distributions in general are just odd. In the left one the red line goes up to 800 m (~2/3 of the landslide), however in the DEM figure both lines are not too far separated. So the differences in the maximum value are quite striking. In the right figure both lines for the distribution have a maximum value of 400 (which is ~20 % of the landslide). Do they belong together at all?
Furthermore, the median values do not make sense. The median should be where the peak of the distribution is. It is especially visible in the left lower panel. I´m not sure why the red line is so much farther to the left. Please clarify!

However, I foremost have trouble understanding the concept of those two distributions in general. Why do you not just take the difference between river and the channel axis? Why involve the landslides at all? Especially considering that you than would just have one value for every landslide.

One side question: In this section I just remembered that you talked in line 208-211 about the fact those slope instabilities do not show any deposits. So how can they change channel morphology if they do not have deposits?

So the planform channel offset DL is the amplitude of your frequency?

L252: This sentence do not make sense to me. Please clarify.

L255: So it is a landslide-induced offset you want to calculate? This should be stated somewhere.

L255: What is a half-wavelength of a landslide? Please describe why you need this parameter.

L256: I wouldn´t call it "flow distance". This is regularly used for the length of flows (like debris, earth or mud flows). Maybe something like "extension of the landslide along the stream".

L265-268: This section can be mentioned in the discussion but not in the methods. Start with "We can use…"

L279: I haven´t found the value for K in both of the references. The first one doesn´t mention the parameter and the second is an AGU talk, with only an abstract available. Please consider updating the references.

L288: So, do you assume that the landslide caused the knickpoints or the knickpoints caused the landslide? If the latter is the case, what caused the knickpoint in the first place?

L292: You also have to consider time! Not only stream power but also with more time, the channel will adapt and straiten again. How do you involve this in your concept?

Results

L299-303: You can put this section either into the methods or in the introduction

L314ff: So do all knickpoints have their origin in landslides or do they develop due to different lithological units? It would be easier to understand if in one map the lithology would be included.

L318: You have never defined ksn before (you defined ks). Please explain what it is.

L319-324: You say that the domain with the highest knickpoint concentration the ksn is also highest (8.1 $m^{0.9}$). Than you say that in the High Himalaya the ksn is lower. However you mention a higher value for Paro (14.7 $m^{0.9}$).

L334: Why do you suddenly make the distinction that you only take landslides which have a certain distance to the channel? You have to explain why! Also consider shifting that sentence to the Methods.

L338: Why not take the hand measurements in the first place?

L339: Why would it be expected?

L341: There are two options which haven´t really been clarified for causing a landslide. a) A knickpoint created by a seismic event, slowly migrates through the stream network, steepening walls and causing a landslide b) a landslide which was caused by rainfall or a seismic event created a knickpoint. If you assume a) than how did the knickpoints develop if there was nearly no seismic activity (lithological knickpoints do not migrate). If b) is the case, how can you be so sure that, considering you are speaking about creeping landslides, one landslides hasn´t produced multiple knickpoints?

L346: A rate, same as frequency, is something that always considers time. So your unit seems odd. This should be something like m/yr. If you want to have a normalization then there is no unit at all.

Discussion

In this section you tend to include information that can be put into the Introduction, Methods, and Results. The main objective, to discuss your results, is barely done. Furthermore, it is obvious that references are missing here (there are none in the discussion). Part of the discussion is to compare and integrate you research into previous work. This needs to be reassessed in detail.

L358: So three geomorphic domains? Not five as explained in line 172?

L359-361: The description of the domains can be put in the Study Area/Introduction.

L361: What are those glacial feature? Moraines? Glacial-fluvial terraces? And what does it signify that the "correlation of fluvial and hillslope activity is less clear" for your study? This might be an interesting point, but you have to discuss it in detail and compare with previous work.

L363-364: You are speaking again about catastrophic failures and not creeping landslides. How do knickpoints affect creeping landslides in general? That would be interesting to discuss. Also, a reference for that statement would be nice.

L364-366: "We calculated…" is a typical beginning for a paragraph in the Method section.

L366-373: "We find…" is a typical beginning for a paragraph in the Result section. Why not put it there? What is your goal in the first place to calculate the density of knickpoints and landslide? Explain (in the Method).

L373-376: You haven´t shown that landslides were just triggered by knickpoint retreat (what about other factors?). It is quite a bold assumption. However, this might be an interesting point which can be discussed further in detail (maybe checking excess topography, lithology influences or climatic changes?).

377: What is AV?

L380-382: You have data about the activity of landslides. It would be interesting to know, not only assume, how the data reflect your statement.

L384-390: Again this section reflect results. You are not discussing anything here.

L390: What do you consider as "characteristic processes within the domain"?

L396-397: Again, how can you assume that knickpoints caused landslides without checking other trigger mechanisms? Fluvial undercutting might be an explanation, as well as strong rainfall.

L397: Why not mention that you take "landslides with a minimum distance of 50 to the river channel" in the Methods? You mention it in the Introduction as well as in the Discussion where it does not belong.

L399: How can you decouple natural fluctuations of the river with an induced channel offset? Are the offsets larger with landslides nearby? This might be something to discuss.

L403: How can the Northern Valleys be a younger domain? What do you mean exactly with this statement? I would assume that uplift started earlier in the Northern Domains. You also mentioned that excess topography as well as knickpoint density are higher in the Northern Valleys. Especially you mention that it has "more time" – so it must be older?! Please explain.

L411: How much more additional time?

L412: What kind of displacement? Channel? Landslide?

L416: I think you "methodological artifacts" are produced by your channel axis crossing hillslope (as visible in Figure 6). This needs to be reevaluated! And also discussed more in detail.

L419-421: The landslide is not a thing that moved during your analysis. So the landside protrudes the same for both the axis as well as for the river channel. Like I said, negative values are produced by axis not following the river channel but crossing the adjacent hillslopes.

 L426: Why normalized time?

L426-430: Again you describe your methods.

L438: What "plot"? A figure?

L442: What can be reasons for "Increased differential stresses"?

L445 ff: This sounds like a conclusion and you already have the first sentence in the conclusion (line 470)! Please remove this paragraph.

Conclusion

L451: "Potentially" creeping. This is new. You have never mentioned it before. So they are not all creeping?

L461-463: What do you mean with "make a case"? Also this sentence is in general hard to understand. What is your point you want to make? Also what are "well-connected domains"? Which domains exactly?

L467: Where do the 300 ka come from? You never mention them before.

L470: "Perhaps" quantify? Either you did it or you don´t.

L474: How could your method support hazard assessment? Might be something to discuss.

Figures

Figure 1: Add small letters on top of the drawings

Figure 2: Why did you add excess topography to the map? What is the additional value? Could you maybe add the geology to the map or the geomorphic domains (Line 166 ff and line 173 ff)? Longitude values should be put underneath or above the corresponding line (It´s therefore hard to read if the lower line is 60°N or 80°N)

Figure 3: Check that no numbers are overlapping with any lines. You can also make the little map bigger.

Figure 4: Mark the exemplary tributary with another color. It is nearly not visible. Also try to stretch the legend more. There is a lot of white space in between.

Figure 6: I would recommend making both the lines and the text thicker and bigger. Furthermore, in the figure description the source is missing ("hillshade derived from….")

Figure 9: The bubbles for "area" in the legend are overlapping each other. Also the colors for "unlikely" and "likely" are hard to distinguish.

Figure 13: The bubble sizes within the plot do not correspond to the bubble size of the legend. Also the bubbles in the legend are overlapping each other.

Technical corrections

1. Remove unnecessary little words such as: "Regularly", "Likely" "in order"

(Examples: L32: "typical", L34: "likely", L35: "in this context", L46: "in order", L52: "ultimately",

L55: "consistently", L38: "this provides additional insight", L390: "the fact that" – it either is or is not. You do not have to specify it as "a fact" (…))

2. Keep it simple

(Examples: L36: Remove "by Incorporating information regarding", L53: Remove "many of the aspects regarding", L59: Replace "have been shown to be capable of broadly reproducing" with "show", L97: Just say that landslides are more likely in excess topography, L255-260: This is a very complicated way to say that the extension of the landslide is called the width.)

3. Remove words (grammar)

(L205: Remove "ranges", L206: Remove "as", L365: remove "for the")

4. Typos

L82: and/or ("and / or")

L86: occur ("occurs")

L95: Quantifies ("Quantify")

L217: ~10 km ("~ 10 km")

L222 + L224: 11.9 m DEM ; 5 m ALOS DEM („11.9m" + „5m")

L443: were ("were were")

5. Other technical comments

L53/54: You mention that erosion has an influence on river morphology twice. You can either delete the first section until "evaluated by". Starting the sentence with "Studying" or integrate your "regional" into the first half of the section and delete everything after "profile".

L57: Knickpoints are waterfalls. Again: keep it simple. You are mentioning the same thing twice. Locally increased profile slopes are basically knickpoints or waterfalls. Why is it important to mention "increase in turbulent flow"?

L97: By definition topography "in excess" in termed "excess topography". You made it very clear the section before. You are just repeating yourself here. Start with "Excess erosion is assumed to be…"

L166/167: Why put "Miocene to Pliocene synorogenic deposits" in '…'? Do you quote here? However, there is no reference added.

L170: If you say "metamorphosed" than you do not need the "meta" in metasedimentary rocks or vice versa.

L171: Replace "located" with "delineate"

L206. Set a point after the parenthesis and start a new sentence with "Almost".

L211-220: I´m not sure if this section is really necessary. Because you repeat every step in the following chapters. This could be removed.

L211: Something like "Between" is missing in the first sentence "between creeping landslides and river channels"

L216-220: Split the last sentence of the first paragraph.

L243: You do not just quantify the offset of valley axis but also the local channel offset, since you describe it two lines further. Maybe delete the first part of the sentence until the first comma.

L256-259: There is something wrong with this sentence. The first part until the comma does not fit well. Please check again!

L334: Insert a "is" between "that" and "less"

**References:** Korup, O. (2005). Geomorphic imprint of landslides on alpine river systems, southwest New Zealand. *Earth Surface Processes and Landforms*, *30*(7), 783-800.

---

## Referee Comment (RC2) · Laure Guerit (Referee) · 9 Feb 2021

The authors propose to develop a measurement of the lateral deviation of river channel due to creeping landslides. They work on a previously published catalogue of creeping landslides in Bhutan and they show that most of these landslides are associated with rivers that deviate from a Âńnormal␞ path. To go further, they propose that creeping landslides are associated with migrating knickpoints that are able to destabilise hillslopes. In addition to the landslide offset, they thus estimate an age since the knickpoint has passed and build a rate of channel offset which they interpret in terms of landslide/channel dynamics through time. The approach is interesting and

supported by very clean and elegant figures. I think it is of interest for publication but the manuscript requires thorough revisions. I do not see any major issue with these revisions so I'm confident the authors will be able to address them. First of all, I think that the manuscript should be reorganized as currently, all the sections are mixed. The introduction is extremely long and does not present clearly the context. The geological setting must be expanded and should be a separate section to ease the reading. More importantly, a lot of results are within the discussion, which does not really discuss the results. This is a bit confusing and it makes the manuscript quite difficult to follow. As a consequence, it is difficult to get a clear idea of the results. Second, some working hypothesis must be better explain and/or justify. For example, the authors mention that tectonic activity is very low and that it can not be responsible for the landslides. They thus associate landslides with migrating knickpoints. But then, what is driving these knickpoints ? And what about climate and lithology ? I identify other aspects that must be clarified in the attached annoted document. Third, I have some issues with the units and names used by the authors. A striking one is the designation of delta chi, a measure in meter, as a time. This is detailed in the attached document. Finally, there are a lot of vague terms like process, low, insights, and of multiple ways to say the same thing (for example comment on Figure 5). Please check for consistency and simplify as much as possible. I also notice minor corrections to do (typo, missing information, missing captions, etc). Again, it prevents me from getting a clear idea of your objectives and results so I really suggest the authors to check the manuscript carefully and to be more explicit to gain in clarity and strength.

Please also note the supplement to this comment:
https://esurf.copernicus.org/preprints/esurf-2020-85/esurf-2020-85-RC2-supplement.pdf

**Supplement:**

[revised manuscript text omitted]

---

## Author Comment (AC1) · 21 Apr 2021

**Planform river channel perturbations resulting from active landsliding in the High Himalaya of Bhutan**

Author response to comments from both referees Larissa de Palézieux, Kerry Leith, Simon Loew

**Content**

| 1. Response to comments from Anonymous Referee #1
2. Response to comments from Laure Guérit | 2
8 |
|------------------------------------------------------------------------------------------------|--------|
|                                                                                                |        |
| References                                                                                     | 14     |

**Color code**

black - original comment from referee blue - author response green - text which will be added to the revised manuscript

**1. Response to comments from Anonymous Referee #1**

The manuscript presented by Palézieux et al. deals with the quantification of lateral channel migration induced by large creeping landslides. They used Fourier transform to separate natural channel amplitudes from landslides-produced variations. Using the inventory of Dini et al. (2020), they show that a large majority of creeping landslides cause lateral channel migration. Furthermore, they hypothesize that creeping landslides are primarily triggered by migrating knickpoints. Under this assumptions they back-calculate the timing of the onset of creeping landslides showing that a phase of activity alternated with a phase of slower movement/inactivity.

Lateral channel migration produced by creeping landslides is not very well quantified and the usage of Fourier transform a somewhat unique approach. Combined with the approach to constraint rates of activity the manuscript shows enough originality to be published.

**1.1. Thank you for taking the time to read our manuscript. Your detailed comments have helped us improve a number of key aspects of the manuscript.**

However, the authors should modify, clarify and/or reassess a couple of points. The manuscript contains a few assumptions which have to be discussed more in detail. The authors mention that seismic activity in the study area is low. This ob- servation is explained in the introduction. Hence, seismic activity is not responsible for triggering creeping landslides. So the authors assume now that knickpoint migration is mainly responsible for triggering creeping landslides. But what about hydrological and climatic influences? What about lithological weak point? Furthermore, migrating knickpoints are primary produced by seismic activity. How do the authors explain the knickpoints in the first place? Especially influences of climate as well as lithology have to be discussed more in detail.

1.2. Thank you. We agree that additional insight into the potential impact of regional geology and tectonics on knickpoint and landslide formation is warranted. However, we feel the discussion on landslide triggering is not relevant to this study. We plan to address these two topics in the following additions to the text (L165):

"The collision of the Indian plate with Asia generates uplift rates of 2-12 mm/yr (Burbank et al., 1996; Wesnousky, 1999, Lavé and Avouac, 2001) and shortening rates of 20-23 mm/yr (Lavé and Avouac, 2000; Burgess et al., 2012) along the Himalayan Arc. Geodetic measurements indicate there is little to no aseismic slip or creep along the Arc, suggesting the boundary is characterized by a strongly coupled fault system in which accumulated stresses are periodically released by large earthquakes (Stevens and Avouac, 2016). Recent studies suggested a several hundred year 'information gap' rather than a 'seismic gap' to be the culprit of the apparent seismic quiescence at the latitudes of Bhutan (Berthet et al., 2014; Hetényi et al., 2016). Two major events have been recorded in the region, one around 1100 A.D. with a magnitude close to M9 (Lavé et al., 2005; Kumar et al., 2010) and in 1714 A.D. with an estimated magnitude of M7.5–8.5 (Hetényi et al., 2016; Berthet et al., 2014; Le Roux-Mallouf et al., 2016). Results from previous authors indicate that the majority of horizontal shortening is accommodated by the Main Frontal Thrust (MFT) (Vernant et al., 2014), a finding supported by observations of fault offsets of several meters recorded in fluvial terraces on the MFT at the latitudes of Bhutan (Berthet et al., 2014; Le Roux-Mallouf

et al., 2016). Long et al. (2019) and Adams et al. (2013) include a number of normal faults on geological maps covering our study region, suggesting a possible switch from compressional to transtensional, or extensional tectonics between the MFT and the northern border of Bhutan. Together, these insights indicate that our study area is likely to be subject to earthquake-induced ground accelerations that may trigger small- to medium-sized landslides, and temporarily reactivate larger slope failures. However, the majority of convergent strains are accumulated on the MFT, and as such, it is unlikely that knickpoints observed in our study region are the result of emergence of local thrust faults."

(R1: L288, L314, R2:L80, 172, 315) We address knickpoint formation and propagation in this study region in a subsequent manuscript. In that study we evaluate the potential for climatic drivers to affect knickpoint formation, with uplift dominating topographic evolution during cool, dry climatic intervals, and knickpoint retreat dominating during warmer, humid periods. This is similar to that identified in the European Alps by Petit et al. (2017), and Leith et al. (2018). We use methods from river profile analysis in combination with field observations derived from large sedimentary plains in the center of the study region to estimate the location of maximum fluvial incision at the end of the current interglacial. We find the maximum upstream location of interglacial fluvial incision is projected to remain within the alluvial infill of the Inner Valley within the Wong Chhu basin, while erosion in the Puna Tsang Chhu basin likely propagates past the alluvial plains, driving baselevel fall in the Northern Valleys. We will include a more general discussion of potential climatic drivers in the revised manuscript.

1.3. To illustrate the spatial relationship of the knickpoints with respect to the lithological boundaries, we have added a geological map (Fig. 1) based on Long et al. (2011), to which we add the boundaries of the geomorphic domains (based on Norbu et al. (2003)), as well as the river network and the knickpoints we adopt in this study. We will add the following text to the discussion (R1:L314, R2: Fig 3):

"While several knickpoints are located within 4 - 5 km of the lower boundary of the South Tibetan Detachment System (STDS), which marks the contact between the Tethyan Himalayan Zone and the Greater Himalayan Zone, the apparent spatial association is likely accentuated by the relatively flat geometry of the STDS at these locations (the elevation of the contact at all knickpoint locations varies by less than 300 m over a contact length of 30 km) (Gansser, 1983, Long et al., 2011). The sedimentary units of the Chekha Formation overlying the STDS are indicated to have both a lower metamorphic grade than the underlying metasediments, with peak temperatures reaching between 500 - 600 °C, while the underlying Greater Himalayan units likely reached 700 - 800 °C (see Long et al. 2019) and references therein). Combined with a significant increase in shear strain indicators in rocks above the SDTS (Long et al., 2019), the rocks in the region of, and structurally overlying the detachment (i.e. downstream of the contact). As such, any knickpoints passing across the contact may be expected to diffuse as erosion accelerates upstream, rather than be stalled and accentuated on the boundary."

1.4. The large scale of the landslides discussed in this manuscript means that they can be considered to be insensitive to triggering by seismic, climatic, or hydrological influences. Such features are almost always associated with large-scale (10's - 100's of meters), long-term (10 - 100 kyr) landscape perturbations (e.g. Agliardi, 2013), and while once initiated, displacement rates may vary with e.g. groundwater levels, a persistent long-term change will be required to alter the boundary conditions (e.g. permanently raise the groundwater table in a >1 Mm3 landslide) enough to generate an observable geomorphic response using our methods. We will add the following text to the introduction (R1: L86, 88/89, 92, 363, R2: L150, L411):

"In this study, we investigate the co-evolution of large creeping landslides and major river channels in NW Bhutan. The scale of the landslides addressed in this study have areas ranging between approximately 0.03 km2 and 3 km2, and as such, can be considered typical of large slope instabilities observed in mountainous environments (Crosta et al. 2013; Agliardi et al., 2013). Such large landslides have been shown to modify topography at the scale of the mountain flank, and shear displacements commonly extend to depths of several hundred meters (Hungr et al., 2014; Bonzanigo et al., 2007; Agliardi et al., 2013; Handwerger et al., 2021). While little absolute dating evidence exists, it is generally accepted that the initiation of many such features in the European Alps pre-date the last glacial maximum (e.g. Tibaldi et al., 2004, Leith et al., 2018), while the onset of the present-day interglacial likely both reactivated existing large features, and led to the development of new ones in response to changes in e.g. valley floor elevation, confining stress, thermal regime, and groundwater levels (e.g. Graemiger et al., 2020, Riva, 2018, Leith et al., 2018, McColl, 2013). Present-day displacement rates for such large landslides are difficult to constrain, as displacement fields tend to be strongly heterogeneous, and while local observations of rates can be on the order of 10's of mm/yr (e.g. Teshebaeva, 2019), it is more common to find displacement rates on mapped large landslides at, or below the accuracy of modern observation techniques (~ 1 mm/yr, e.g. Dini, 2019a). Combined with morphological observations, monitoring of such slope instabilities indicate that displacements can best be characterized by a combination of long-lasting activity, episodic reactivations, and/or continuous slow creep (e.g. Crosta et al. 2013). Accelerations up to rapid velocities (m/hour, (Hungr et. al., 2014)) or greater may be expected to eventually lead to catastrophic failure, however, based on morphological and sedimentary evidence we do not expect many, if any of the landslides in our dataset have achieved such rates, and more likely creep at similar rates since their formation.

Numerical models seeking to track the development of large landslides in similar settings typically implement plastic constitutive models that combine a prescribed progressive strength reduction (attempting to capture the progressive weathering of a rock mass), with a shear-strain weakening of frictional and cohesive properties (representing mechanical rock mass damage). Although such boundary conditions will ultimately lead to global slope failure as material strength is allowed to reduce below that required to maintain topographic stresses, associated failures will tend to propagate from the most highly loaded slope toe, to the crest. This results in both an anomalously deep sliding surface, and a short interval between initiation of damage at the toe and global slope failure, which is inconsistent with observations of in-situ stepping landslides in the landscape today. Implementing a progressive removal of mass within the associated valley, either through the removal of a glacier load (e.g. Riva et al., 2018), or erosion of bedrock in the valley floor (e.g. Hou et al.,

2014), has the effect of addressing both of these issues by progressively loading hillslope elements close to the upper surface of the buttress, and effectively migrating a damage zone down the hillslope. Such elasto-plastic constitutive models typically adopt model steps as a proxy for time, modulated by the rate of prescribed buttress removal, or strength degradation (e.g. Spreafico et al., 2020), and as such cannot truly capture time-dependent behaviour (including the onset of failure, and creep rates). Nonetheless, these observations that such large landslides require a) progressive weathering, b) strain-dependant damage, and c) progressive buttress removal indicate that isolated contributions from climatic or groundwater changes, weak lithology, or seismic activity cannot explain the presence of landslides discussed in this manuscript."

Another point are the methods, which seems to contain a few flaws. Especially when calculating the planform channel offset DL. The calculated channel axis, probably derived from the DEM, is crossing the hillslope (Figure 6). Resulting in negative values in the distance distribution of the landslides. Especially the distance distributions (Figure 6 bottom) pose some questions. Please, check the comments in added in the supplement file.

1.5. We realize that we have not clearly addressed certain properties of the distance calculations. We have modified figure 6 (Fig. 2 in this document) to better illustrate the two types of distance distribution and we will rephrase the corresponding lines as follows (R1:L243, L22-230, L338, R2: L221, L416, L335, L419, L243, L335, L416):

"We evaluate the amplitude of river channel sinuosity (Dva), and landslide impingement into the valley (D1) in a reference system described in terms of the valley axis. In this case, the x-coordinate of the landform (channel or landslide) is determined by the upstream distance from the rangefront, while the y-coordinate is determined with respect to the valley axis (e.g., positive on the true left of valleys, negative on the true right). Since the Fourier transform measures the quantity of each frequency component in terms of peak-to-peak amplitudes, the absolute position of the zero-datum has no impact on our derived amplitudes of sinuosity. To calculate landslide impingement, we first derive the median distance to the river channel by reprojecting our landslide features in a coordinate system relative to the river channel (Figure 2). We then compare this to the median distance derived from the reference valley axis coordinate system, each normalized by the minimum distance to the channel, and the y-position of the lowest cell in the landslide, respectively, in order to align the toe of the landslide (assumed to be the lower 1/3 of the landslide mass) in the two coordinate systems. For cases in which D1 is positive, the curvature of the landslide toe with respect to the valley axis exceeds the amplitude of channel sinuosity with respect to the landslide toe (i.e. the landslide offset dominates the channel form), while zero, or negative values of D1 indicate the landslide toe is relatively straight with respect to the channel, and the river has likely abandoned the landslide toe. This metric allows us to directly compare the amplitude of sinuosity derived from the Fourier transform, with the apparent landslide impingement described by the complex 3D geometry of the landslide toe, without users having to manually locate features on the landslide, for example, the upstream and downstream corners, or the point of maximum displacement (Figure 2). We recognise this is a complicated approach, however, as quantifying this relationship is entirely novel we feel it is important to provide an objective measure to consistently capture this hillslope - channel interaction within a range of mountainous landscapes."

Especially the discussion needs further improvement since a lot of the written paragraphs belong to the Methods or Results. Rarely any of the Results are put into perspective with previous research.

1.6. We will reorganize the methods and results sections, and include a more in-depth comparison of our results with those presented in previous research. In particular:

- We present a new method to specifically evaluate the amplitude of channel sinuosity. We will discuss this with reference to previous methods to characterize path length sinuosity (e.g. Tarboton et al., 1988), and physical controls on planform curvature (e.g. Stark et al., 2006).
- (L93): While we believe our assumption that knickpoint migration leads to the initiation of large creeping landslides is intuitive, few authors have directly investigated such a relationship. We will discuss our findings with respect to fluvial incision-induced landsliding in Taiwan, Japan, and Papua New Guinea (Tsou et al. 2014, Hou et al., 2014, Hovius et al., 1998), and associated present-day relief in the Himalaya (Blöthe et al., 2015, Korup et al., 2010).
- We will add the following discussion (L25): "Aside from a limited study by Othman and Gloaguen (2013), we believe this is the only study to quantitatively evaluate the long-term interaction of fluvial channel morphology with deep-seated creeping landslides. Notably, Korup (2005) suggested that deep-seated creeping landslides can lead to "diversion of channels around deposits, causing incision of meandering gorges.", while Korup (2006) noted that an increase in weathering and secondary instability on large landslides tends to produce more subdued hillslope morphology, and may reduce rates of long-term fluvial incision. Korup et al. (2006) were able to confirm these hypotheses, at least over Holocene timescales, by quantifying the effects of river blocking rock slides on the position, and cross-sectional geometry of adjacent river channels, however, these observations are limited to events that fail dynamically, and the implications for longer timescales, or creeping instabilities are not clear. Our results therefore contribute new tools to extend the current state-of-the-art, and aid the quantification of both landscape evolution, and large creeping landslide activity from geomorphological datasets."
- We will include the following discussion (R1: L373): "Our unique approach allows us to suggest (with some assumptions) the initiation of the oldest landslides currently observed within our study area may occur soon after the MPT. Although this is consistent with the findings of Korup and Schlunegger (2007), who suggest giant landslides may be features of mountain belts at all stages of evolution, the assumed ages presented in this manuscript are significantly greater than those currently reported in literature, which, to date either constrain maximum ages as post-LGM (e.g., McCalpin, 1995, Agliardi et al., 2013), the last interglacial (e.g. Tibaldi et al., 2004, Baroň et al., 2013), or in a unique case in the Polish Tatra mountains 280 ka (Szczygieł et al, 2019). Constraining the timing of deep-seated landslide initiation, or long-term displacement history is, however, notoriously difficult, as heterogeneity due to (for example) parasitic slope failures cause problems for the interpretation of dating evidence. In addition their position in active mountain belts means that glacial activity commonly overprints any evidence for displacement prior to the LGM. We see some correlation between the density estimate for landslide ages, and timing of marine isotope stages (Figure 3), indicating landslide initiation may be at least partly controlled by long-term climatic influences. By providing evidence to suggest that the

initiation of some large creeping rock slope instabilities may exceed most previous estimates by an order of magnitude, we hope to inspire new approaches to better constrain conditions controlling the onset of instability in such environments (e.g., by numerical modelling, improved geomorphological analyses, or dating of speleothems).

Regarding the general structure: The introduction is too long (10 pages). Therefore, I would recommend to revise how necessary certain explanations of methods are (e.g. lateral channel migration is explained in the Introduction as well as in the Methods) and if there is the possibility to add another section "Regional setting".

1.7. We will restructure the introduction, remove redundant sections (e.g., L56, 114), and move sections relevant to the technical aspects of our study to the methods section. We will create a separate section characterizing the study area and will focus the geological description more closely on NW Bhutan (see above 1.3). In line with your comments to line 314 and figure 2 we have added a geological map showing the spatial relationship between the lithological contacts, geomorphic domains, glacial overprint, river network, and knickpoints (Fig. 2).

Regarding the sentence structure and usage of scientific terminology: Even though methods are explained in quite the detail, often terminology is used without proper definition and in varying contexts. Examples are mentioned in the technical details. Furthermore, sentence structures are often too long and too complicated which results in grammar mistakes, sometimes leading to sentences which are hard to understand.

1.8. We will reformulate the methods section to be more precise with respect to terminology, units, and equations and will adapt the terminology throughout the manuscript accordingly.

**2. Response to comments by Laure Guérit**

The authors propose to develop a measurement of the lateral deviation of river channel due to creeping landslides. They work on a previously published catalogue of creeping landslides in Bhutan and they show that most of these landslides are associated with rivers that deviate from a 'normal' path. To go further, they propose that creeping landslides are associated with migrating knickpoints that are able to destabilise hillslopes. In addition to the landslide offset, they thus estimate an age since the knickpoint has passed and build a rate of channel offset which they interpret in terms of landslide/channel dynamics through time.

**2.1. Great summary, thanks!**

The approach is interesting and supported by very clean and elegant figures. I think it is of interest for publication but the manuscript requires thorough revisions. I do not see any major issue with these revisions so I'm confident the authors will be able to address them.

2.2. Thank you very much for your input. We hope the following responses support your interpretation. Thank you also for your constructive and detailed comments. We agree with your remarks regarding the structure, terminology, and precision of language of our manuscript and have noticed that we have not well presented our working hypotheses and line of reasoning.

First of all, I think that the manuscript should be reorganized as currently, all the sections are mixed. The introduction is extremely long and does not present clearly the context. The geological setting must be expanded and should be a separate section to ease the reading.

2.3. (R2:L45, L58) We aim to restructure the introduction such as to detail our assumptions and hypotheses thoroughly and concisely (see also paragraphs below). We will remove redundant paragraphs/lines (e.g. L56, 114) in order to present only the necessary previous research without anticipating information belonging to the methods chapter. We will move the characterization of the study area to a separate section following the introduction and focus the geological description more closely on NW Bhutan. In this context we will also add a geological map (Fig. 1) with the river network and the outlines of the geomorphic domains and add the geomorphic domains as suggested to the long profiles.

This is similar to a comment by Reviewer 1. for further details, please see our response 1.3.

In line with your comments we will homogenize the methods section with respect to the level of detail with which each method is presented (e.g., L110), remove repetitions (e.g., L261), and correct the commented equations, units, and terminology (e.g. Fig. 5, L244, 251, 254, 255, 272 (Eq. 13), 278).

More importantly, a lot of results are within the discussion, which does not really discuss the results. This is a bit confusing and it makes the manuscript quite difficult to follow. As a consequence, it is difficult to get a clear idea of the results.

2.4. (R2: L356, L396) In the results section we will expand the pointed-out statements (e.g. lines 307-312 and 316) and include those parts of the discussion, which are more suited to be presented at this location. We will rephrase and restructure the paragraphs for flow of reading and clarity of argumentation. We agree that a thorough discussion of our results and particularly the integration with respect to previous research is lacking and will rewrite the chapter accordingly. We will reformulate the conclusions to be more explicit and coherent where commented and to be in line with the discussion.

This is similar to a comment by Reviewer 1. For further detail, please see Response 1.6.

Second, some working hypothesis must be better explain and/or justify. For example, the authors mention that tectonic activity is very low and that it can not be responsible for the landslides. They thus associate landslides with migrating knickpoints. But then, what is driving these knickpoints ? And what about climate and lithology ? I identify other aspects that must be clarified in the attached annoted document.

2.5. We agree that we have not addressed our assumptions regarding the origin and type of knickpoints in our study area in detail (comments to lines 150, 172, and 396). We will add additional text to address this in the revised manuscript:

This is similar to a comment by Reviewer 1. For further detail, please see response 1.2.

Third, I have some issues with the units and names used by the authors. A striking one is the designation of delta chi, a measure in meter, as a time. This is detailed in the attached document.

2.6. We had previously discussed this point, and felt 'normalized' would help readers unfamiliar with chi plots grasp the concepts we present. However, at your suggestion, we have changed our terminology for referring to chi from 'normalized time' to 'interval'. We have also corrected equations 13 and 14, where the units should now be correct (R1: L346):

Eq. 13:  $\tau = \frac{\chi}{KA_0^m} \chi$ with K: erodibility [m0.9 a-1], A0 the reference drainage area, m = 0.45, and n = 1.

Eq. 14:  $R = \frac{D_L}{\Delta \tau} = \frac{KA_0^m D_L}{\Delta \chi}$ with DL: planform channel deviations [m].

We have noticed that there is an error in the terminology of the different distance distributions, which we will correct in the revised manuscript. We have also updated figure 6 (Fig. 2 in this document) to better illustrate the difference between the distance distributions with respect to the original channel and with respect to the valley axis.

Finally, there are a lot of vague terms like process, low, insights, and of multiple ways to say the same thing (for example comment on Figure 5). Please check for consistency and simplify as much as possible. I also notice minor corrections to do (typo, missing information, missing captions, etc). Again, it prevents me from getting a clear idea of your objectives and

results so I really suggest the authors to check the manuscript carefully and to be more explicite to gain in clarity and strength.

2.7. We have added two paragraphs to the introduction of our manuscript, which we would like to present here to better explain our working hypotheses regarding the formation and long-term evolution of large creeping landslides with respect to the knickpoints in our study area (comments to lines 58, 80, 123, 150, 172, and 441).

This is similar to a comment by Reviewer 1. For further detail, please see Response 1.4.

**Figures**

---

## Author Comment (AC2) · 21 Apr 2021

Thank you very much for your thoughtful and helpful comments.

Please find our combined response to the comments of both referees in the supplementary document.

Please also note the supplement to this comment:
https://esurf.copernicus.org/preprints/esurf-2020-85/esurf-2020-85-AC2-supplement.pdf
**ESurfD**

Interactive
comment